# Semi-Relaxed Quantization with DropBits: Training Low-Bit Neural Networks via Bit-wise Regularization

## Abstract

Network quantization, which aims to reduce the bit-lengths of the network weights and activations, has emerged as one of the key ingredients to reduce the size of neural networks for their deployments to resource-limited devices. In order to overcome the nature of transforming continuous activations and weights to discrete ones, recent study called Relaxed Quantization (RQ) [Louizos et al. 2019] successfully employ the popular Gumbel-Softmax that allows this transformation with efficient gradient-based optimization. However, RQ with this Gumbel-Softmax relaxation still suffers from large quantization error due to the high temperature for low variance of gradients, hence showing suboptimal performance. To resolve the issue, we propose a novel method, *Semi-Relaxed Quantization (SRQ)* that uses multi-class straight-through estimator to effectively reduce the quantization error, along with a new regularization technique, *DropBits* that replaces dropout regularization to randomly drop the bits instead of neurons to reduce the distribution bias of the multi-class straight-through estimator in SRQ. As a natural extension of DropBits, we further introduce the way of learning heterogeneous quantization levels to find proper bit-length for each layer using DropBits. We experimentally validate our method on various benchmark datasets and network architectures, and also support a new hypothesis for quantization: learning heterogeneous quantization levels outperforms the case using the same but fixed quantization levels from scratch.

## 1 Introduction

Deep neural networks have achieved great success in various computer vision applications such as image classification, object detection/segmentation, pose estimation, action recognition, and so on. However, state-of-the-art neural network architectures require too much computation and memory to be deployed to resource-limited devices. Therefore, researchers have been exploring various approaches to compress deep neural networks to reduce their memory usage and computation cost.

In this paper, we focus on *neural network quantization*, which aims to reduce the bit-width of a neural network while maintaining competitive performance with a full-precision network. It is typically divided into two groups, uniform and heterogeneous quantization. In uniform quantization, one of the simplest methods is to round the full-precision weights and activations to the nearest grid points: $\widehat{x} = \alpha \lfloor \frac{x}{\alpha} + \frac{1}{2} \rfloor$ where $\alpha$ controls the grid interval size. However, this naïve approach incurs severe performance degradation on large datasets. Recent network quantization methods tackle this problem from different perspectives. In particular, Relaxed Quantization (RQ) (Louizos et al., 2019) employs Gumbel-Softmax (Jang et al., 2017; Maddison et al., 2017) to force weights and activations to be located near quantization grids with high density. Louizos et al. (2019) notice the importance of keeping the gradient variance small, which leads them to use high Gumbel-Softmax temperatures in RQ. However, such high temperatures may cause a large quantization error, thus preventing quantized networks from achieving comparable performance to full-precision networks.

To resolve this issue, we first propose *Semi-Relaxed Quantization (SRQ)* that uses the mode of the original categorical distribution in the forward pass, which induces small quantization error. It is clearly distinguished from Gumbel-Softmax choosing $argmax$ among the *samples* from the concrete distribution. To cluster weights cohesively around quantization grid points, we devise a multi-class straight-through estimator (STE) that allows for efficient gradient-based optimization as well. As this STE is biased like a traditional one (Bengio et al., 2013) for the binary case, we present a novel technique, *DropBits* to reduce the distribution bias of the multi-calss STE in SRQ. Motivated from Dropout (Srivastava et al., 2014), DropBits drops bits rather than neurons/filters to train low-bit neural networks under SRQ framework.

In addition to uniform quantization, DropBits allows for *heterogeneous quantization*, which learns different bit-width per parameter/channel/layer by dropping redundant bits. DropBits with learnable bit-drop rates adaptively finds out the optimal bit-width for each group of parameters, possibly further reducing the overall bits. In contrast to recent studies (Wang et al., 2019; Uhlich et al., 2020) in heterogeneous quantization that exhibit almost all layers possess *at least* 4 bits, up to 10-bit, our method yields much more resource-efficient low-bit neural networks with *at most* 4 bits for all layers.

With trainable bit-widths, we also articulate a *new hypothesis for quantization* where one can find the learned bit-width network (termed a 'quantized sub-network') which can perform better than the network with the same but fixed bit-widths from scratch.

Our contribution is threefold:

- We propose a new quantization method, **S**emi-**R**elaxed **Q**uantization (SRQ) that introduces the multi-class straight-through estimator to reduce the quantization error of Relaxed Quantization for transforming continuous activations and weights to discrete ones. We further present a novel technique, **DropBits** to reduce the distribution bias of the multi-class straight-through estimator in SRQ.

- Extending DropBits technique, we propose a more resource-efficient heterogeneous quantization algorithm to curtail redundant bit-widths across groups of weights and/or activations (e.g. across layers) and verify that our method is able to find out 'quantized sub-networks'.

- We conduct extensive experiments on several benchmark datasets to demonstrate the effectiveness of our method. We accomplish new **state-of-the-art** results for ResNet-18 and MobileNetV2 on the ImageNet dataset when *all* layers are uniformly quantized.

## 2 RELATED WORK

While our goal in this work is to obtain an extremely low-bit neural network both for weights and activations, here we broadly discuss existing quantization techniques with various goals and settings. BinaryConnect (Courbariaux et al., 2015) first attempted to binarize weights to $\pm 1$ by employing deterministic or stochastic operation. To obtain better performance, various studies (Rastegari et al., 2016; Li et al., 2016; Achterhold et al., 2018; Shayer et al., 2018) have been conducted in binarization and ternarization. To reduce hardware cost for inference, Geng et al. (2019) proposed softmax approximation via a look-up table. Although these works effectively decrease the model size and raise the accuracy, they are limited to quantizing weights with activations remained in full-precision. To take full advantage of quantization at run-time, it is necessary to quantize activations as well.

Researchers have recently focused more on simultaneously quantizing both weights and activations (Zhou et al., 2016; Yin et al., 2018; Choi et al., 2018; Zhang et al., 2018; Gong et al., 2019; Jung et al., 2019; Esser et al., 2020). XNOR-Net (Rastegari et al., 2016), the beginning of this line of work, exploits the efficiency of XNOR and bit-counting operations. QIL (Jung et al., 2019) also quantizes weights and activations by introducing parametrized learnable quantizers that can be trained jointly with weight parameters. Esser et al. (2020) recently presented a simple technique to approximate the gradients with respect to the grid interval size to improve QIL. Nevertheless, these methods do not quantize the first or last layer, which leaves a room to improve power-efficiency.

For ease of deployment in practice, it is inevitable to quantize weights and activations of all layers, which is the most challenging. Louizos et al. (2019) proposed to cluster weights by using Gumbel-Softmax, but it shows drawbacks as we will discuss in Section 3.2. Jain et al. (2019) presented efficient fixed-point implementations by formulating the grid interval size to the power of two, but they quantized the first and last layer to at least 8-bit. Zhao et al. (2020) proposed to quantize the grid interval size and network parameters in batch normalization for the deployment of quantized models on low-bit integer hardware, but it requires a specific accelerator only for this approach.

As another line of work, Fromm et al. (2018) proposed a heterogeneous binarization given pre-defined bit-distribution. HAWQ (Dong et al., 2019) determines the bit-width for each block heuristically based on the top eigenvalue of Hessian. Unfortunately, both of them do not learn optimal bit-widths for heterogeneity. Toward this, Wang et al. (2019) and Uhlich et al. (2020) proposed a layer-wise heterogeneous quantization by exploiting reinforcement learning and learning dynamic range of quantizers, respectively. However, their results exhibit that almost all layers have up to 10-bit (at least 4-bit), which would be suboptimal. Lou et al. (2020) presented a channel-wise heterogeneous quantization by exploiting hierarchical reinforcement learning, but channel-wise precision limits the structure of accelerators, thereby restricting the applicability of the model.

## 3 METHOD

In this section, we briefly review Relaxed Quantization (RQ) (Louizos et al., 2019) and propose Semi-Relaxed Quantization, which selects the nearest grid point in the forward pass to decrease the quantization error. To make it learnable and to cluster compressed parameters cohesively, SRQ expresses the nearest grid selection of the forward pass as the equivalent form, the combination of logistic distribution and argmax, and performs the backward pass on it. Then, we present DropBits technique to reduce the distribution bias of SRQ and its extension to heterogeneous quantization.

### 3.1 PRELIMINARY: RELAXED QUANTIZATION

Relaxed Quantization (RQ) considers the following quantization grids for weights: $\widehat{\mathcal{G}} = \alpha[-2^{b-1}, \ldots, 0, \ldots, 2^{b-1} - 1] =: [g_0, \ldots, g_{2^b-1}]$ where $b$ is the bit-width and a learnable parameter $\alpha > 0$ for each layer controls a grid interval. When quantizing activations, the grid points in $\widehat{\mathcal{G}}$ start from zero since the output of ReLU is always non-negative. Then, $x$ (a weight or an activation) is perturbed by noise $\epsilon$ as $\widetilde{x} = x + \epsilon$, which enables gradient-based optimization for non-differentiable rounding. The noise $\epsilon$ follows a distribution $p(\epsilon) = \text{Logistic}(0, \sigma)$ so that $p(\widetilde{x})$ is governed by $\text{Logistic}(x, \sigma)$ where $\sigma$ represents the standard deviation. Under $p(\widetilde{x})$, we can easily compute the *unnormalized* probability of $\widetilde{x}$ being quantized to each grid point $g_i$ in a closed form as below:

$$\pi_i = p(\widehat{x} = g_i | x, \alpha, \sigma) = \text{Sigmoid}\big((g_i + \alpha/2 - x)/\sigma\big) - \text{Sigmoid}\big((g_i - \alpha/2 - x)/\sigma\big), \quad (1)$$

where $\widehat{x}$ denotes a quantized realization of $\widetilde{x}$. Note that the cumulative distribution function of the logistic distribution is just a sigmoid function. Finally, given unnormalized categorical probability $\boldsymbol{\pi} = \{\pi_i\}_{i=0}^{2^b-1}$ for grid points $\widehat{\mathcal{G}} = \{g_i\}_{i=0}^{2^b-1}$, RQ discretizes $x$ to $\widehat{x} = \boldsymbol{r} \cdot \widehat{\mathcal{G}}$ by sampling $\boldsymbol{r} = \{r_i\}_{i=0}^{2^b-1}$ from the concrete distribution (Jang et al., 2017; Maddison et al., 2017) with a temperature $\tau$:

$$u_i \sim \text{Gumbel}(0, 1), \quad r_i = \frac{\exp\big((\log \pi_i + u_i)/\tau\big)}{\sum_{j=0}^{2^b-1} \exp\big((\log \pi_j + u_j)/\tau\big)}, \quad \widehat{x} = \sum_{i=0}^{2^b-1} r_i g_i. \quad (2)$$

The algorithm of RQ is described in Appendix in detail.

### 3.2 SEMI-RELAXED QUANTIZATION - FIXING PITFALLS OF RQ

Although RQ achieves competitive performance with both weights and activations of neural networks quantized, the quantization probability modeling of RQ may still incur large quantization error, thereby yielding suboptimal performance. To be specific, Louizos et al. (2019) recommend high temperatures for the concrete distribution (e.g. $\tau = 1.0$ or $2.0$) in (2) since exploiting low temperatures hinders networks from converging due to high variance of gradients. However, it turns out that

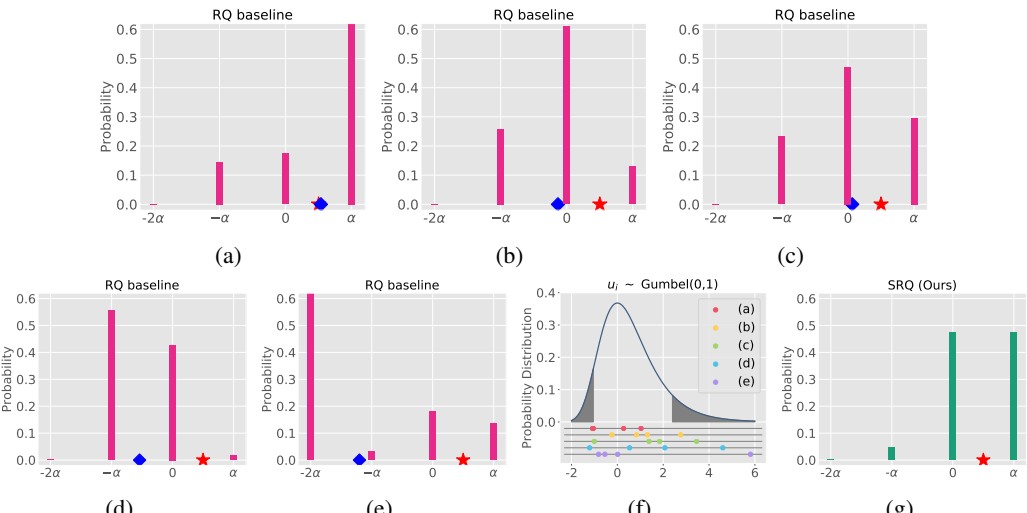

Figure 1: Probability of $\star$ being quantized to each grid point by RQ ($\tau = 1.0$) and SRQ respectively. (a $\sim$ e): RQ according to *different* Gumbel samples, (f): Gumbel samples used in (a $\sim$ e), and (g): SRQ, the same as the original categorical distribution. The $x$-axis denotes weight value (except (f)), $\star$ denotes the original value $\alpha/2$. For RQ (a$\sim$e), $\blacklozenge$ represents $\widehat{x}$ for input $x = \star$ computed by (2).

the concrete distribution with such a high temperature is almost similar to the uniform distribution. As a concrete example, we consider 2-bit quantization with $\widehat{\mathcal{G}} = \alpha[-2, -1, 0, 1]$ for a fixed scale parameter $\alpha > 0$, $\sigma = \alpha/3$, and we set $\tau$ to 1.0 as in Louizos et al. (2019). Now, suppose that the original weight value is $\alpha/2$. As in Figure 1-(b,d,e), $\star$ can be sporadically quantized to below zero by RQ as the original categorical distribution has support for $-\alpha$ and $-2\alpha$. It may be okay on average, but RQ computes only one sample in each forward pass due to computational burden, which can accidentally lead to very large quantization error for these particular sample.

To avoid the counterintuitive-sample with large quantization error as seen in Figure 1-(b,d,e), we propose 'Semi-Relaxed Quantization' (SRQ) which rather directly considers the original categorical distribution in Figure 1-(g). To be concrete, for a weight or an activation $x$, the probability of $x$ being quantized to each grid is $r_i = \pi_i / \sum_{j=0}^{2^b-1} \pi_j$ for $i \in \{0, \cdots, 2^b - 1\}$ with $b$-bit precision, where $\pi_i$ is computed as (1). In such a manner, selecting a grid point for $x$ can be thought of as sampling from the categorical distribution with categories $\widehat{\mathcal{G}} = \{g_i\}_{i=0}^{2^b-1}$ and the corresponding probabilities $\mathbf{r} = \{r_i\}_{i=0}^{2^b-1}$ as illustrated in Figure 1-(g). Then, the grid point $g_{i_{max}}$ with $i_{max} = \text{argmax}_i r_i$ would be the most reasonable speculation due to the highest probability. SRQ therefore chooses the mode of the original categorical distribution, $g_{i_{max}}$ and assign it to $\widehat{x}$, entirely discriminated from Gumbel-Softmax which selects the argmax among samples from the concrete distribution. As a result, SRQ does not suffer from counterintuitive-sample problem that RQ encounters at all.

The last essential part for SRQ is to handle the non-differentiable argmax operator in computing $i_{max}$. Toward this, we propose a multi-class straight-through estimator (STE) that allows for backpropagating through a non-differentiable *categorical* sample by approximating $\partial\mathcal{L}/\partial r_{i_{max}}$ to $\partial\mathcal{L}/\partial y_{i_{max}}$ and $\partial\mathcal{L}/\partial r_i$ to zero for $i \neq i_{max}$, where $\mathcal{L}$ is the cross entropy between the true label and the prediction made by a quantized neural network as delineated in the previous paragraph and $y_{i_{max}}$ is the $i_{max}$-th entry of the one-hot vector $y$. The forward and backward passes of SRQ are summarized as follows.

**Forward:** $y = \texttt{one\_hot}[\underset{i}{\text{argmax}}\, r_i]$, **Backward:** $\dfrac{\partial\mathcal{L}}{\partial r_{i_{max}}} = \dfrac{\partial\mathcal{L}}{\partial y_{i_{max}}}$ and $\dfrac{\partial\mathcal{L}}{\partial r_i} = 0$ for $i \neq i_{max}$ (3)

Such a formulation brings two important advantages in network quantization. First of all, (3) makes the variance of gradient estimator become zero. Since SRQ always chooses the mode of the original categorical distribution (i.e., there is no randomness in the forward pass of SRQ), and the gradient of loss function $\mathcal{L}$ with respect to the individual categorical probabilities is defined as *zero* everywhere except for the coordinate corresponding to the mode, the variance of gradients in SRQ is indeed zero.

The other advantage is that the backward pass (3) can cluster network weight parameters cohesively. Under the assumption that $r_i = \pi_i$, $\frac{\partial\mathcal{L}}{\partial x}$ is proportional to

Figure 2: Weight distributions for 3-bit quantized LeNet-5 by (a) RQ and (b) SRQ. The $x$-axis and $y$-axis represent the weight values and their frequencies, respectively. The vertical dashed lines denote grid points.

$\frac{\partial \pi_{i_{\max}}}{\partial x} = \frac{1}{\sigma}\big(\text{Sigmoid}\big(\frac{g_{i_{\max}}+\alpha/2-x}{\sigma}\big)\text{Sigmoid}\big(-\frac{g_{i_{\max}}+\alpha/2-x}{\sigma}\big) - \text{Sigmoid}\big(\frac{g_{i_{\max}}-\alpha/2-x}{\sigma}\big)\text{Sigmoid}\big(-\frac{g_{i_{\max}}-\alpha/2-x}{\sigma}\big)\big)$[1]. When $x$ is close to $g_{i_{\max}}$, $\partial \pi_{i_{\max}}/\partial x$ is nearly zero, so is $\partial \mathcal{L}/\partial x$. With an appropriate learning rate, $x$ converges to $g_{i_{\max}}$, which leads SRQ to cluster weights better than RQ as shown in Figure 2. Although $\frac{\partial \mathcal{L}}{\partial x}$ is almost zero, $\alpha$ is still trained. After $\alpha$ is updated, there is a gap between $x$ and $\alpha$ so that $x$ can be trained. Hence, the network will continue to train until it finds the optimal $\alpha$.

### 3.3 DROPBITS

Although our multi-class STE enjoys low variance of gradients, it is biased to the mode as the binary one in Bengio et al. (2013). To reduce the bias of a STE, Chung et al. (2016) propose the slope annealing trick, but this strategy is only applicable to the binary case. To address this limitation, we propose a novel method, *DropBits*, to decrease the distribution bias of a multi-class STE. Inspired by dropping neurons in Dropout (Srivastava et al., 2014), we *drop an arbitrary number of grid points at random every iteration*, where in effect the probability of being quantized to dropped grid points becomes zero.

However, the design policy that each grid point has its own binary mask would make the number of masks increase exponentially with bit-width. Taking into account appropriate noise levels with a less aggressive design, the following two examples are available: **(a)** endpoints in the grids share the same binary mask, and **(b)** the grid points in the same bit-level share the same binary mask (see Figure 3). Hereafter, we consider **(b)** bitwise-sharing masks for groups of grid points, unless otherwise specified.

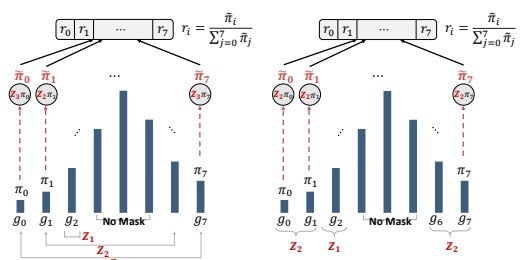

(a) Endpoints-sharing mask  (b) Bitwise-sharing mask

Figure 3: Two mask designs in 3-bit

Now, we introduce how to formulate binary masks. Unlike practical Dropout implementation through dividing activations by $1 - p$ (here, $p$ is a dropout probability), we employ an explicit binary mask $Z$ whose probability $\Pi$ can be optimized jointly with model parameters. The Bernoulli random variable being non-differentiable, we relax a binary mask via the *hard concrete* distribution (Louizos et al., 2018). While the binary concrete distribution (Maddison et al., 2017) has its support $(0, 1)$, the hard concrete distribution stretches it slightly at both ends, thus concentrating more mass on exact 0 and 1. Assuming disjoint masks, we describe the construction of a binary mask $Z_k$ for the $k$-th bit-level using the hard concrete distribution as

$$U_k \sim \text{Uniform}(0, 1), \ S_k = \text{Sigmoid}\Big(\big(\log U_k - \log(1 - U_k) + \log \Pi_k - \log(1 - \Pi_k)\big)/\tau'\Big) \quad (4)$$

$$\bar{S}_k = S_k(\zeta - \gamma) + \gamma \quad \text{and} \quad Z_k = \min(\max(\bar{S}_k, 0), 1)$$

where $\tau'$ is a temperature for the hard concrete distributions with $\gamma < 0$ and $\zeta > 0$ reflecting stretching level. For $i = 2^{b-1} - 1, 2^{b-1}$ and $2^{b-1} + 1$, we do not sample from the above procedure but fix $Z = 1$ so as to prohibit all the binary masks from becoming zero (see 'No Mask' in Figure 3).

With the value of each mask generated from the above procedure, the probability of being quantized to any grid point is re-calculated by multiplying $\pi_i$'s by their corresponding binary masks $Z_k$'s (e.g. $\widetilde{\pi}_0 = Z_2 \cdot \pi_0$) and then normalizing them (to sum to 1). As seen in Figure 4, the sampling distribution

---

[1]Since $i_{max}$ does not change, the assumption is not unreasonable.

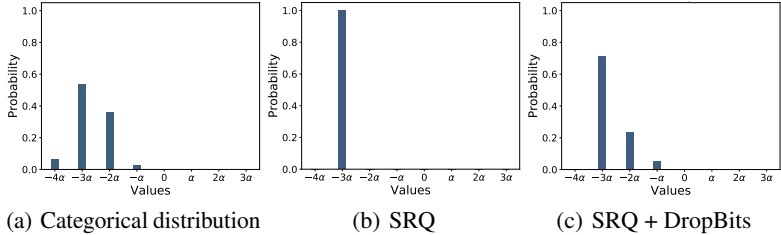

(a) Categorical distribution       (b) SRQ       (c) SRQ + DropBits

Figure 4: The illustration of the effect of DropBits on SRQ. For a certain weight, **(a)** the categorical distribution indicates $r_i = \pi_i / \Sigma_{j=0}^{7} \pi_j$ for each grid $(i = 0, \cdots, 7)$, **(b)** the distribution of SRQ is a sampling distribution after taking the argmax of $r_i = \pi_i / \Sigma_{j=0}^{7} \pi_j$, and **(c)** the distribution of SRQ + DropBits is a sampling distribution after taking the argmax of $r_i = \widetilde{\pi}_i / \Sigma_{j=0}^{7} \widetilde{\pi}_j$. Here, $\Pi_k$'s are initialized to 0.7 for clear understanding.

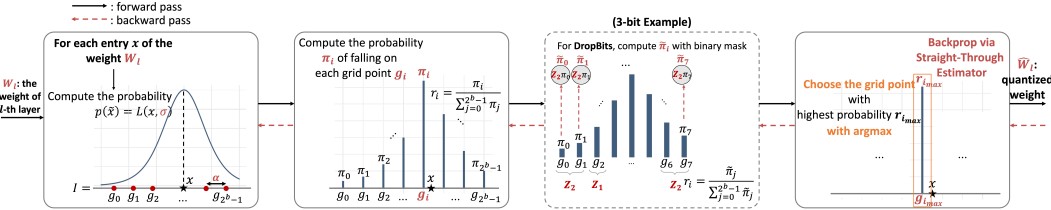

Figure 5: Illustration of Semi-Relaxed Quantization (SRQ) framework with DropBits technique.

of SRQ is biased to the mode, $-3\alpha$. With DropBits adjusting $\pi_i$'s to $\widetilde{\pi}_i$'s based on $Z_k$'s, the sampling distribution of SRQ + DropBits more resemble the original categorical distribution than that of SRQ, which means that DropBits could effectively reduce the distribution bias of the multi-class straight-through estimator in SRQ. Not only that, DropBits does not require any hand-crafted scheduling at all due to the learnable characteristic of $\Pi_k$, whereas such scheduling is vital for Gumbel-Softmax (Jang et al., 2017; Maddison et al., 2017) and slope annealing trick (Chung et al., 2016).

Although quantization grids for weights are symmetric with respect to zero, those for activations start from zero, which makes it difficult to exploit symmetrically-designed DropBits for activations. Therefore, DropBits is applied only for weights in our experiments. Assuming that $Z_k$'s are shared across all weights of each layer, the overall procedure is described in Figure 5. The overall algorithm including the test phase is deferred to Appendix due to space limit.

### 3.4 EMPIRICAL ANALYSIS OF QUANTIZATION ERROR, DISTRIBUTION BIAS, VARIANCE

In this section, we empirically compare (i) the expectation of quantization error, (ii) the $\ell_2$-norm of distribution bias, and (iii) the gradient variance of RQ, SRQ, and SRQ + DropBits. The expectation of quantization error indicates the expected value of difference between an input value and its quantized value, $\mathbb{E}[|x - \widehat{x}|]$ where $\widehat{x}$ is a quantized value of $x$ by each algorithm. Here, the expectation is taken over the randomness of $\widehat{x}$.

As described in Appendix B, it is not possible to compute the bias of gradient with respect to $\pi$, so we instead compute the bias of gradient with respect to $x$ as a proxy. However, it is not straightforward to say that SRQ + DropBits is better than RQ and vice versa in terms of the bias with respect to $x$ (see Figure 8-(b) in Appendix B). As an additional indirect metric, we suggest a distribution bias as the difference between the original categorical distribution and the distribution approximated by each algorithm, $\mathbb{E}[\boldsymbol{p}] - \boldsymbol{\pi}_{\text{origin}}$ where a vector $\boldsymbol{\pi}_{\text{origin}} := (\pi_i / \sum_{i=0}^{2^b-1} \pi_j)_{i=0}^{2^b-1}$ is the original categorical distribution and a vector $\boldsymbol{p} := (p_i)_{i=0}^{2^b-1}$ is the approximated one (i.e., $(p_i)_{i=0}^{2^b-1} = (r_i)_{i=0}^{2^b-1}$ in (2) for RQ, and $(p_i)_{i=0}^{2^b-1} = (y_i)_{i=0}^{2^b-1}$ in (3) for SRQ (+ DropBits)). As a distribution bias is also a vector, we compare the $\ell_2$-norm of a distribution bias of each algorithm. Note that our new notion of distribution bias is very loosely coupled with the gradient in the sense that both converge to zero if there is no approximation for the non-differentiable parts. However, it can be used as an indirect indicator to measure how much biased distribution is used in computing the gradient estimator.

For the variance, we use the closed-form of the gradient estimator $\frac{\partial r}{\partial \pi}$ of each algorithm: for $i \neq j$, (a) RQ: $\frac{\partial r_i}{\partial \pi_j} = -\frac{r_i r_j}{\tau \pi_j}$ with randomness on $u_i$ and $u_j$, and (b) SRQ + DropBits: $\frac{\partial r_i}{\partial \pi_j} = -\frac{r_i r_j}{\pi_j}$ with randomness on binary masks $Z_i$ and $Z_j$. The case for $i = j$ can be derived similarly.

Figure 6: Comparison of RQ, SRQ, and SRQ + DropBits in quantization error/distribution bias/variance.

Armed with these terms, we conduct a case study in 3-bit quantization with grid points $\widehat{\mathcal{G}} = \alpha[-4, \cdots, 3]$ via Monte Carlo simulation when $\alpha$ is simply set to $1.0$. Here, let $x$ be the midpoint of consecutive grid points, i.e., $x = -3.5\alpha, \cdots, 2.5\alpha$. For gradient variance, since there are many pairs of $\frac{\partial r_i}{\partial \pi_j}$ for different $i$ and $j$, we representatively choose $i$ as the index of the grid point closest to $x$, and $j$ as the indices of two neighboring grid points of $x$ (e.g. if $x \in [g_{i-1}, g_i]$, then $j = i - 1$ and $i$).

In Figure 6, SRQ shows smaller quantization error than RQ for all $x = -3.5\alpha, \cdots, 2.5\alpha$. This is because SRQ deliberately performs biased estimation on the underlying categorical distribution to prevent large quantization error from even occurring in the forward pass while sharing this underlying distribution with RQ in the backward pass. Instead, we devised DropBits to reduce the incurred distribution bias of SRQ, which is indeed the case as can be seen in Figure 6. Interestingly, SRQ + DropBits can also achieve smaller distribution bias than RQ for all $x = -3.5\alpha, \cdots, 2.5\alpha$.

## 3.5 LEARNING BIT-WIDTH TOWARDS RESOURCE-EFFICIENCY

As noted in Section 1 and 2, recent studies on heterogeneous quantization use at least 4-bit in almost all layers, up to 10-bit, which leaves much room for the saving of energy and memory. Towards more resource-efficient one, we introduce an additional regularization on DropBits to drop redundant bits.

Since the mask design in Figure 3-(b) reflects the actual bit-level and the probability of each binary mask in DropBits is learnable, we can penalize the case where we use higher bit-levels via a sparsity encouraging regularizer like $\ell_1$. As Louizos et al. (2018) proposed a relaxed $\ell_0$ regularization using the hard concrete binary mask, we adopt this continuous version of $\ell_0$ as a sparsity inducing regularizer. Following (4), we define the smoothed $\ell_0$-norm as $\mathcal{R}(Z; \Pi) = \text{Sigmoid}(\log \frac{\Pi}{1-\Pi} - \tau' \log \frac{-\gamma}{\zeta})$. One caveat here is that we do not have to regularize masks for low bit-level if a higher bit-level is still alive (in this case such a high bit-level is still necessary for quantization). We thus design a regularization in such a specific way as only to permit the probability of a binary mask at the current highest bit-level to approach zero. More concretely, for bit-level binary masks $\{Z_k\}_{k=1}^{b-1}$ as in Figure 3-(b) and the corresponding probabilities $\{\Pi_k\}_{k=1}^{b-1}$, our regularization term to learn the bit-width is

$$\mathcal{R}\big(\{Z_k\}_{k=1}^{b-1}, \{\Pi_k\}_{k=1}^{b-1}\big) = \sum_{k=1}^{b-1} \mathbb{I}(Z_k > 0)\Big(\prod_{j=k+1}^{b-1} \mathbb{I}(Z_j = 0)\Big)\mathcal{R}(Z_k; \Pi_k).$$

Note that $\{Z_k\}_{k=1}^{b-1}$ is assigned to each group (e.g. all weights or activations in a layer or channel for instance). Hence, every weight in a group shares the same sparsity pattern (and bit-width as a result), and learned bit-widths across groups are allowed to be heterogeneous.

Assuming the $l$-th *layer* shares binary masks $\boldsymbol{Z}^l \coloneqq \{Z_k^l\}_{k=1}^{b-1}$ associated with probabilities $\boldsymbol{\Pi}^l \coloneqq \{\Pi_k^l\}_{k=1}^{b-1}$, our final objective function for a $L$-layer neural network becomes $\mathcal{L}(\boldsymbol{\theta}, \boldsymbol{\alpha}, \boldsymbol{\sigma}, \boldsymbol{Z}, \boldsymbol{\Pi}) + \lambda \sum_{l=1}^{L} \mathcal{R}(\boldsymbol{Z}^l, \boldsymbol{\Pi}^l)$, where $\boldsymbol{\alpha} = \{\alpha_l\}_{l=1}^{L}$ and $\boldsymbol{\sigma} = \{\sigma_l\}_{l=1}^{L}$ represent the layer-wise grid interval parameters and standard deviations of logistic distributions, $\boldsymbol{Z} = \{\boldsymbol{Z}^l\}_{l=1}^{L}$, $\boldsymbol{\Pi} = \{\boldsymbol{\Pi}^l\}_{l=1}^{L}$, and $\lambda$ is a regularization parameter. In inference phase, we just drop unnecessary bits based on the values of $\boldsymbol{\Pi}$.

## 3.6 NEW HYPOTHESIS FOR QUANTIZATION

Frankle & Carbin (2019) articulated the 'lottery ticket hypothesis', stating that one can find some sparse sub-networks, 'winning tickets', from randomly-initialized, dense neural networks that are easier to train than sparse networks resulting from pruning. In this section, we define a new hypothesis for quantization with slightly different (opposite in some sense) perspective from the original one.

**Notation.** $a \succ_{\text{bit}} b$ and $a =_{\text{bit}} b$ denote that $a$ has strictly higher bit-width than $b$ for at least one of all groups (e.g. channels or layers), and $a$ has the same bit-precision as $b$ across all groups, respectively.

**Definition.** For a network $f(x; \theta)$ with randomly-initialized parameters $\theta$, we consider a quantized network $f(x; \theta')$ from $f(x; \theta)$ such that $\theta \succ_{\text{bit}} \theta'$. If the accuracy of $f(x; \theta')$ is higher than that

Table 1: Test error (%) for LeNet-5 on MNIST and VGG-7 on CIFAR-10. "Ann." stands for annealing the temperature $\tau$ in RQ.

| Dataset | # Bits W./A. | RQ | RQ + Ann.[2] | SRQ | SRQ + DropBits |
|---------|------|-----|---------|-----|---------|
| MNIST | 4/4 | 0.58 | 0.62 | 0.59 | **0.53** |
| | 3/3 | 0.69 | 0.74 | 0.67 | **0.58** |
| | 2/2 | 0.76 | − | 0.72 | **0.63** |
| CIFAR-10 | 4/4 | 8.43 | 8.47 | 7.15 | **6.85** |
| | 3/3 | 9.56 | 10.78 | 7.08 | **6.94** |
| | 2/2 | 11.75 | − | 7.68 | **7.51** |

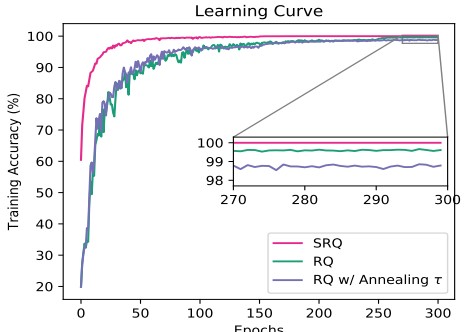

Figure 7: Learning curves of VGG-7 quantized by RQ, RQ with annealing $\tau$, and SRQ in 3-bit.

Table 2: Top-1/Top-5 error (%) with ResNet-18 and MobileNetV2 on ImageNet.

| Method | # Bits W./A. | ResNet-18 Top-1/Top-5 | MobileNetV2 Top-1/Top-5 |
|--------|------|-----------------|-------------------|
| Full-precision | 32/32 | 30.24/10.92 | 28.12/9.71 |
| RQ (Louizos et al., 2019) | 4/4 | 38.48/16.01 | −/− |
| RQ ST (Louizos et al., 2019) | 4/4 | 37.54 / 15.22 | − / − |
| QIL[3] (Jung et al., 2019) | 4/4 | 31.05/11.23 | 32.77/12.51 |
| LLSQF (Zhao et al., 2020) | 4/4 | 30.60/11.28 | 32.63/12.01 |
| | 3/3 | 33.33/12.58 | −/− |
| TQT (Jain et al., 2019; Uhlich et al., 2020) | 4/4 | 30.49/− | 32.21/− |
| **SRQ + DropBits** | 4/4 | **30.37/10.96** | **30.83/11.26** |
| | 3/3 | **32.79/12.57** | **36.96/15.20** |

of $f(x; \theta'')$ where $f(x; \theta'')$ is trained from scratch with fixed bit-widths such that $\theta' =_{\text{bit}} \theta''$, then $f(x; \theta')$ is referred to as a *quantized sub-network* of $f(x; \theta)$.

This hypothesis implies that learning bit-width would be superior to pre-defined bit-width. To the best of our knowledge, our study is the first attempt to delve into this hypothesis.

## 4 EXPERIMENTS

Since popular deep learning libraries such as TensorFlow (Abadi et al., 2016) and PyTorch from v1.3 (Paszke et al., 2019) already provide their own 8-bit quantization functionalities, we focus on lower bit-width regimes (i.e. $2 \sim 4$-bit). In contrast to some other quantization papers, our method uniformly quantizes the weights and activations of *all* layers containing both the *first* and *last* layers. We first show that SRQ and DropBits have its own contribution, none of which is negligible. Then, we evaluate our method, SRQ + DropBits on a totally large-scale dataset with deep networks. Finally, we demonstrate that our heterogeneous quantization method yields promising results even if all layers have at most 4-bit and validate a new hypothesis for quantization in Section 3.6.

### 4.1 ABLATION STUDIES

To validate the efficacy of SRQ and DropBits, we successively apply each piece of our method to RQ for LeNet-5 (LeCun et al., 1998) on MNIST and VGG-7 (Simonyan & Zisserman, 2014) on CIFAR-10. Table 1 shows that SRQ outperforms RQ in most cases. One might wonder that the issue of RQ introduced in Section 3.2 can be addressed by an annealing schedule of the temperature $\tau$ in RQ. It could be possible, but RQ with an annealing schedule suffers from high variance of gradients due to low temperatures at the end of training as shown in Figure 7. As a result, annealing $\tau$ gives rise to worse performance than RQ as shown in Table 1. However, SRQ does not suffer from both problems at all, thus displaying the best learning curve in Figure 7. Finally, it can be clearly identified that DropBits consistently improves SRQ by decreasing the distribution bias of our multi-class STE.

---

[2]We cannot reproduce the results of RQ in 2-bit, so we experiment only on 3-bit and 4-bit RQ

[3]Our own implementation with all layers quantized by using pretrained models available from PyTorch

Table 3: Test error (%) for quantized sub-networks using LeNet-5 on MNIST, VGG-7 on CIFAR-10,and ResNet-18 on ImageNet. Here, an underline means the learned bit-width and "T" stands for ternary precision.

| Model | Initial # Bits W/A | Test Error | Trained W. Bits per layer | Test Error (Fixed) | Test Error (Reg.) |
|---|---|---|---|---|---|
| LeNet-5 | 4/4 | 0.53 | 4/4/$\underline{3}$/4 | 0.55 | **0.52** |
|  | 3/3 | 0.58 | 3/$\underline{2}$/3/3 | 0.65 | **0.55** |
|  | 2/2 | 0.63 | 2/2/2/$\underline{T}$ | 0.68 | **0.59** |
| VGG-7 | 4/4 | 6.77 | 4/4/4/4/4/$\underline{3}$/4 | 6.74 | **6.65** |
|  | 3/3 | 6.82 | 3/3/3/3/3/$\underline{2}$/3/3 | 6.81 | **6.77** |
|  | 2/2 | 7.49 | 2/2/2/2/2/2/2/$\underline{T}$ | 7.43 | **7.36** |
| ResNet-18 | 4/4 | 33.20 | 4/$\underline{3}$/$\underline{3}$/$\underline{3}$/$\underline{3}$/$\underline{3}$/$\underline{3}$/$\underline{3}$/$\underline{3}$/$\underline{3}$/$\underline{3}$/$\underline{3}$/$\underline{3}$/4/4/$\underline{3}$/4/4/4 | 34.58 | **34.30** |
|  | 3/3 | 37.80 | 3/3/$\underline{2}$/3/$\underline{2}$/3/3/3/3/3/3/$\underline{2}$/3/3/3/3/3/3/3 | 41.01 | **40.30** |

### 4.2 RESNET-18 AND MOBILENETV2 ON IMAGENET

To verify the effectiveness of our algorithm on the ImageNet dataset, we quantize the ResNet-18 (He et al., 2016) and MobileNetV2 (Sandler et al., 2018) architectures initialized with each pre-trained full-precision network available from the official PyTorch repository. In Table 2, our method is only compared to the state-of-the-art algorithms that quantize both weights and activations of *all* layers for fair comparisons. The extensive comparison against recent works that remain the first or last layer in the full-precision is given in Appendix.

Table 2 illustrates how much better our model performs than the latest quantization methods as well as our baseline, RQ. In ResNet-18, SRQ + DropBits outdoes RQ, QIL, LLSQF, and TQT, even achieving the top-1 and top-5 errors in 4-bit nearly close to those of the full-precision network. In MobileNetV2, SRQ + DropBits with 4-bit surpasses all existing studies by more than one percentage point. Moreover, we quantize MobileNetV2 to 3-bit, obtaining competitive performance, which is remarkable due to the fact that none of previous works successfully quantizes MobileNetV2 to 3-bit.

### 4.3 FINDING QUANTIZED SUB-NETWORKS

In this experiment, we validate a *new hypothesis for quantization* by training the probabilities of binary masks using the regularizer in Section 3.5 to learn the bit-width of each layer. For brevity, only weights are heterogeneously quantized, and the bit-width for activations is fixed to the initial one.

In Table 3, the fourth column represents the bit-width per layer learned by our regularizer, and the fifth and last columns indicate the test error when fixing the bit-width of each layer same as trained bit-widths (fourth column) from scratch and when using our regularization approach, respectively. Table 3 shows that a learned structure by our heterogeneous quantization method (last column) is superior to the fixed structure with learned bit-widths from scratch (fifth column) for all cases. It might be doubtful whether our regularizer is able to recognize which layer is really redundant or not. This may be indirectly substantiated by the observation that the fixed structure with trained bit-widths from scratch (fifth column) outperforms the uniform quantization (third column) on CIFAR-10. More experiments on different values of the regularization parameter $\lambda$ are deferred to Appendix .

## 5 CONCLUSION

We proposed *Semi-Relaxed Quantization (SRQ)*, which effectively clusters the weights in low bit-width regimes, along with *DropBits* that reduces the distribution bias of SRQ. We empirically showed that both SRQ and DropBits possess its own value, thereby leading SRQ + DropBits to achieve the state-of-the-art performance for ResNet-18 and MobileNetV2 on ImageNet. Furthermore, we took one step forward to consider heterogeneous quantization by simply penalizing binary masks in DropBits, which enables us to find out quantized sub-networks. As future work, we plan to extend our heterogeneous quantization method to activations and its application to other quantizers.

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

# A    ALGORITHM OF SEMI-RELAXED QUANTIZATION WITH DROPBITS

---

**Algorithm 1** **S**emi-**R**elaxed **Q**uantization (SRQ) + DropBits

---

1: **Input:** Training data
2: **Initialize:** Bit-width $b$, network parameters $\{W_l, b_l\}_{l=1}^L$, layer-wise grid interval parameters and the standard deviations of a logistic distribution in the $l$-th layer $\{\alpha_l, \sigma_l\}_{l=1}^L$. Initialize layer-wise grid $\widehat{\mathcal{G}}_l = \alpha_l[-2^{b-1}, \cdots, 2^{b-1} - 1] =: [g_{l,0}, g_{l,1}, \cdots, g_{l,2^b-1}]$ for $l \in \{1, \cdots, L\}$.
3: **procedure** TRAINING
4:     **for** $l = 1, \cdots, L$ **do**
5:         $x \leftarrow$ Each entry of $W_l$ or $b_l$
6:         $I_l = \widehat{\mathcal{G}}_l - \alpha/2$                                                        ▷ Shift the grid by $-\alpha/2$
7:         $F = \text{Sigmoid}\left(\frac{I_l - x}{\sigma_l}\right)$                                        ▷ Compute CDFs
8:         $\pi_i = F[i+1] - F[i]$    for $i = 0, \cdots, 2^b - 1$
9:         **if** DropBits **then**
10:             Sample a mask $Z_k$ for $k = 0, \cdots, b-1$ from (4)
11:             $\widetilde{\pi} = \pi \odot Z$
12:             $r_i = \widetilde{\pi}_i / \sum_{j=0}^{2^b-1} \widetilde{\pi}_j$                    ▷ Figure 3
13:         **else**
14:             $r_i = \pi_i / \sum_{j=0}^{2^b-1} \pi_j$
15:         **end if**
16:         $y = \texttt{one\_hot}[\text{argmax}_i \, r_i]$                    ▷ Multi-class Straight-Through Estimator
17:         $\widehat{x} = y \odot \widehat{\mathcal{G}}_l$                                                        ▷ Quantization
18:         $\widehat{W}_l \leftarrow$ Each entry of $W_l$ quantized to $\widehat{x}$
19:         $\widehat{b}_l \leftarrow$ Each entry of $b_l$ quantized to $\widehat{x}$

20:         Forward pass with quantized $\widehat{W}_l$ and $\widehat{b}_l$
21:         Activation can be quantized in the same way, with DropBits being False
22:     **end for**
23: **end procedure**
24:
25: **procedure** DEPLOYMENT
26:     **for** $l = 1, \cdots, L$ **do**
27:         $\widehat{W}_l = \min(\max(\alpha_l \cdot \text{Round}(W_l/\alpha_l), g_{l,0}), g_{l,2^b-1})$
28:         $\widehat{b}_l = \min(\max(\alpha_l \cdot \text{Round}(b_l/\alpha_l), g_{l,0}), g_{l,2^b-1})$
29:     **end for**
30: **end procedure**

---

# B COMPARISON OF BIAS BETWEEN RQ, SRQ, AND SRQ + DROPBITS

In general, when the loss involves discrete random variables, the true gradient, the gradient of the expectation of the loss with respect to the parameter of discrete random variables, can be obtained by using existing stochastic gradient estimation techniques such as the score function estimator (see Equation (4) and (5) in Maddison et al. (2017)). However, the reason why we did not explicitly compare the bias of gradient is that distribution parameters (i.e. $\boldsymbol{\pi} = \{\pi_i\}_{i=0}^{2^b-1}$) in our setting are not independent of network parameters $x$. In fact, $\boldsymbol{\pi}$ is a function of $x$ in (1). Given that an inverse of a function $\boldsymbol{\pi}$ does not exist because $\pi_i$ is not one-to-one with respect to $x$ for each $i$, it is not possible to directly apply existing techniques such as the score function to compute unbiased estimators. Instead, we compute the bias of gradient with respect to $x$ as a proxy of that of with respect to $\boldsymbol{\pi}$.

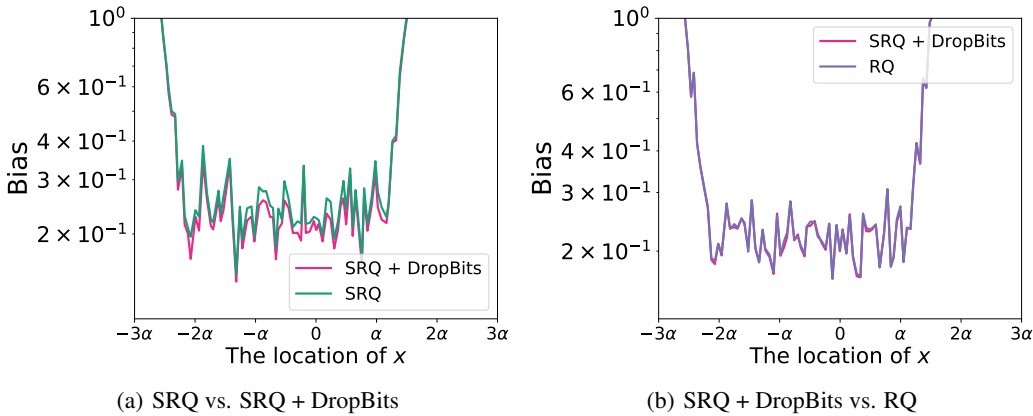

(a) SRQ vs. SRQ + DropBits      (b) SRQ + DropBits vs. RQ

Figure 8: The comparison of bias **(a)** between SRQ and SRQ + DropBits, and **(b)** between SRQ + DropBits and RQ, when training LeNet-5 on MNIST in 3-bit. In this experiment, only weights are quantized. The $x$-axis denotes the value of a weight in the last fully connected layer.

Although the bias of SRQ + DropBits is smaller than that of SRQ as displayed in Figure 8-(a), SRQ + DropBits and RQ exhibit comparable levels of bias with respect to $x$ as shown in Figure 8-(b).

# C  EXTENSIVE COMPARISON FOR RESNET-18 AND MOBILENETV2 ON IMAGENET

Our method, SRQ + DropBits surpasses quantization methods remaining the first or last layer in the full precision as well as the latest algorithms that quantize both the weights and activations of all layers including the first and last layers, except QIL (Jung et al., 2019) and LSQ (Esser et al., 2020) both of which utilize the full-precision first and last layer as well as employ their own ResNet-18 pretrained model performing much higher than one available from the official PyTorch repository.

Table 4: Top-1/Top-5 error (%) with ResNet-18 and MobileNetV2 on ImageNet using 4-bit. † denotes the use of the full-precision first or last layer, and ‡ indicates our own implementation with all layers quantized by using pretrained models available from the official PyTorch repository.

|  | ResNet-18 | | MobileNetV2 | |
|---|---|---|---|---|
| Methods | Top-1 | Top-5 | Top-1 | Top-5 |
| Full-precision | 30.24 | 10.92 | 28.12 | 9.71 |
| DoReFa† (Zhou et al., 2016) | 31.9 | — | — | — |
| BCGD† (Yin et al., 2018) | 32.64 | 12.24 | — | — |
| LQ-Nets† (Zhang et al., 2018) | 30.7 | 11.2 | — | — |
| PACT† (Choi et al., 2018; Wang et al., 2019) | 30.8 | — | 38.56 | 17.30 |
| RQ (Louizos et al., 2019) | 38.48 | 16.01 | — | — |
| RQ ST (Louizos et al., 2019) | 37.54 | 15.22 | — | — |
| DSQ† (Gong et al., 2019) | 30.44 | - | 35.20 | — |
| QIL‡ (Jung et al., 2019) | 31.05 | 11.23 | 32.77 | 12.51 |
| QIL† (Jung et al., 2019) | 29.90 | — | — | — |
| TQT (Jain et al., 2019; Uhlich et al., 2020) | 30.49 | — | 32.21 | — |
| LLSQF (Zhao et al., 2020) | 30.60 | 11.28 | 32.63 | 12.01 |
| LSQ† (Esser et al., 2020) | 28.90 | 10.0 | — | — |
| **SRQ + DropBits (Ours)** | **30.37** | **10.96** | **30.83** | **11.26** |

## D    MORE EXPERIMENTS ON HETEROGENEOUS QUANTIZATION

As we can see in Table 5, our heterogeneous quantization method is capable of finding quantized sub-networks in a broad range of regularization parameter $\lambda$.

Table 5: Test error (%) for quantized sub-networks using LeNet-5 on MNIST, VGG-7 on CIFAR-10,and ResNet-18 on ImageNet. Here, an underline means the learned bit-width and "T" stands for ternary precision.

| Model | Initial # Bits W/A | Test Error | Trained W. Bits per layer | Test Error (Fixed) | Test Error (Reg.) |
|---|---|---|---|---|---|
| LeNet-5 | 4/4 | 0.53 | 4/4/3̲/4 | 0.55 | **0.52** |
| | | | 4/3̲/3̲/4 | 0.59 | **0.58** |
| | 3/3 | 0.58 | 3/2̲/3/3 | 0.65 | **0.55** |
| | | | 3/T̲/3/2̲ | 0.85 | **0.60** |
| | 2/2 | 0.63 | 2/2/2/T̲ | 0.68 | **0.59** |
| | | | T̲/2/2/T̲ | 0.70 | **0.64** |
| VGG-7 | 4/4 | 6.77 | 4/4/4/4/3̲/3̲/4 | 6.74 | **6.65** |
| | | | 4/3̲/4/4/3̲/3̲/4 | 6.87 | **6.80** |
| | 3/3 | 6.82 | 3/3/3/3/2̲/3/3 | 6.81 | **6.77** |
| | | | 3/2̲/2̲/2̲/T̲/2̲/3 | 7.13 | **7.04** |
| | 2/2 | 7.49 | 2/2/2/2/2/2/T̲ | 7.43 | **7.36** |
| | | | T̲/T̲/T̲/T̲/T̲/T̲/2/T̲ | 9.62 | **7.55** |
| ResNet-18 | 4/4 | 33.20 | 4/3̲/3̲/3̲/3̲/3̲/3̲/3̲/3̲/3̲/3̲/3̲/3̲/4/4/3̲/4/4/4 | 34.58 | **34.30** |
| | | | 3̲/3̲/3̲/3̲/3̲/3̲/3̲/3̲/3̲/3̲/3̲/3̲/3̲/3̲/3̲/3̲/3̲/3̲/4 | 36.46 | **34.94** |
| | 3/3 | 37.80 | 3/3/2̲/3/2̲/3/3/3/3/3/3/2̲/3/3/3/3/3/3/3 | 41.01 | **40.30** |
| | | | 3/3/2̲/2̲/2̲/2̲/3/3/3/3/3/3/2̲/3/3/3/3/3/3/3 | 43.41 | **42.13** |

# E  COMPARISON OF SRQ + DROPBITS WITH GUMBEL-SOFTMAX + MULTI-CLASS STE

As described in Section 3.4, our SRQ + DropBits shows smaller quantization error, variance of gradients, and distribution bias than RQ, while maintaining stochasticity. For this reason, we employ the deterministic scheme in the first place and then encourage stochasticity via DropBits.

To show the effectiveness of SRQ + DropBits further, we empirically compare it with an algorithm using the Gumbel-Softmax STE in the forward pass instead of DropBits and our multi-class STE in the backward pass. Let such an algorithm be called "Gumbel-Softmax + multi-class STE".

Table 6: Test error (%) for LeNet-5 on MNIST and VGG-7 on CIFAR-10.

| Dataset | # Bits W./A. | Gumbel-Softmax + multi-class STE | SRQ + DropBits |
|---|---|---|---|
| MNIST | 4/4 | 0.57 | **0.53** |
|  | 3/3 | 0.65 | **0.58** |
|  | 2/2 | 0.74 | **0.63** |
| CIFAR-10 | 4/4 | 8.25 | **6.85** |
|  | 3/3 | 8.44 | **6.94** |
|  | 2/2 | 10.41 | **7.51** |

Although employing our multi-class STE in the backward pass, Gumbel-Softmax + multi-class STE performs worse than SRQ + DropBits. This is primarily due to the fact that Gumbel-Softmax STE still incurs a large quantization error like RQ.

# F    IMPLEMENTATION DETAILS

The weights and activations of all layers including the first and last layers (denoted by $W$ and $A$) are assumed to be perturbed as $\widetilde{W} = W + \epsilon$ and $\widetilde{A} = A + \epsilon$ respectively, under $\epsilon \sim L(0, \sigma)$ as we describe in Section 2.

Concerning DropBits regularization in 3.3, we initialize the probability of each binary mask with $\Pi \sim \mathcal{N}(0.9, 0.01^2)$ (i.e. corresponding to low dropout probability). The concrete distribution of a binary mask is stretched to $\zeta = 1.1$ and $\gamma = -0.1$ as recommended in Louizos et al. (2018), and $\tau'$ is initialized to 0.2 to make a binary mask more discretized.

For MNIST experiments, we train LeNet-5 with 32C5 - MP2 - 64C5 - MP2 - 512FC - Softmax architecture for 100 epochs irrespective of the bit-width. In addition, a learning rate is set to 5e-4 regardless of the bit-width and exponentially decayed with decay factor 0.8 for the last 50 epochs. The input is normalized into $[-1, 1]$ range without any data augmentation.

For CIFAR-10 experiments, following the convention that the location of max-pooling layer is changed, which originates from Rastegari et al. (2016), a max-pooling layer is located after a convolutional layer, but before a batch normalization and an activation function. We train VGG-7 with 2x(128C3) - MP2 - 2x(256C3) - MP2 - 2x(512C3) - MP2 - 1024FC - Softmax architecture for 300 epochs, and a learning rate is initially set to 1e-4 regardless of the bit-width. The learning rate is multiplied by 0.1 at 50% of the total epochs and decay exponentially with the decay factor 0.9 during the last 50 epochs. The input images are preprocessed by substracting its mean and dividing by its standard deviation. The training set is augmented as follows: (i) a random $32 \times 32$ crop is sampled from a padded image with 4 pixels on each side, (ii) images are randomly flipped horizontally. The test set is evaluated without any padding or cropping. Note that a batch normalization layer is put after every convolutional layer in VGG-7, but not in LeNet-5.

In Section 4.1, RQ with an annealing schedule of the temperature $\tau$ in RQ is implemented by following Jang et al. (2017): $\tau$ is annealed every 1000 iterations by the schedule $\tau = \max(0.5, \exp{(-t/100000)})$ in 3-bit and $\tau = \max(0.5, 2\exp{(-t/100000)})$ in 4-bit in order to make the decreasing rate of $\tau$ as small as possible. Here, $t$ is the global training iteration.

For ImageNet experiments in Section 4.2, the weight parameters of both ResNet-18 and MobileNetv2 are initialized with the pre-trained full precision model available from the official PyTorch repository. When quantizing ResNet-18 to 3-bit, fine-tuning is implemented for 80 epochs with a batch size of 256: a learning rate is initialized to 2e-5 and divided by two at 50, 60, and 68 epochs. When quantizing ResNet-18 to 4-bit, fine-tuning is carried out for 150 epochs with a batch size of 128: for the first 125 epochs, a learning rate is set to 5e-6, but 1e-6 for the last 25 epochs. When quantizing MobileNetV2 to 3-bit and 4-bit, fine-tuning is performed for 25 epochs with a batch size of 48 and an initial learning rate of 2e-5: the learning rate is divided by two at 15 and 20 epochs for 3-bit and at 10, 12, 18, and 20 epochs for 4-bit. We employ AdamW in Decoupled Weight Decay Regularization (Loshchilov & Hutter, 2019) with a weight decay factor of 0.01.

In Section 4.3 and D, if the probability of a binary mask is less than 0.5, then we drop the corresponding bits. For LeNet-5 on MNIST and VGG-7 on CIFAR-10, our regularization term in Section 3.5 is activated only for the first 50% of the total epochs. With the remained bit-width for each layer, fine-tuning process is conducted for the last 50% of the total epochs. For ResNet-18 on ImageNet, we initialize the weights of ResNet-18 with the pre-trained full precision model and train it for ten epochs for simplicity. During training, our regularization term in Section 3.5 is activated only for the first 9 epochs, and fine-tuning process is done for the last epoch with the remained bit-width of each layer fixed. All experiments in Table 3 and 5 were conducted by the use of AdamW: the weight decay value is set to 0.01 for LeNet-5, 0.02 for VGG-7, and 0.01 for ResNet-18. We consider the regularization parameter $\lambda \in [5 \times 10^{-5}, 10^{-2}]$ to encourage layer-wise heterogeneity.

# G ALGORITHM OF RQ

We provide the algorithmic details of RQ as follows. For quantizing weights, $\widehat{\mathcal{G}} = [g_i]_{i=0}^{2^b-1} = \alpha[-2^{b-1}, \ldots, 0, \ldots, 2^{b-1} - 1]$; however, when quantizing activations, the grid points start from zero since the outputs of ReLU activations are always non-negative, that is, $\widehat{\mathcal{G}} = [g_i]_{i=0}^{2^b-1} = \alpha[0, \ldots, 2^b - 1]$. The objective function in RQ is cross-entropy loss function of class labels and class probabilities predicted by quantized weights, biases, and activations, which is the same as the loss function $\mathcal{L}$ in our method.

---

**Algorithm 2** Relaxed Quantization (RQ) Louizos et al. (2019) for training

1: **Input:** $x$ (a weight or an activation)
2: **Initialize:** scale $\alpha$, standard deviation $\sigma$, grids $\widehat{\mathcal{G}} = [g_i]_{i=0}^{2^b-1} = [g_0, \cdots, g_{2^b-1}]$
3: **Require:** bit-width $b$, temperature $\tau$
4: $I = \left[ g_0 - \dfrac{\alpha}{2}, \cdots, g_{2^b-1} - \dfrac{\alpha}{2}, g_{2^b-1} + \dfrac{\alpha}{2} \right]$
5: $F = \text{Sigmoid}\left( \dfrac{I - x}{\sigma} \right)$ ▷ Compute CDF
6: $\pi_i = F[i+1] - F[i]$ for $i = 0, \cdots, 2^b - 1$
7: ▷ Unnormalized Prob. for each grid point
8: # Sampling from the concrete distribution
9: $u_i \sim \text{Gumbel}(0, 1)$ for $i = 0, \cdots, 2^b - 1$
10: $r_i = \dfrac{\exp\big((\log \pi_i + u_i)/\tau\big)}{\sum_j \exp\big((\log \pi_j + u_j)/\tau\big)}$
11: **Output:** $\widehat{x} = \sum_{i=0}^{2^b-1} r_i g_i$

---

**Algorithm 3** Relaxed Quantization (RQ) Louizos et al. (2019) for inference

1: **Input:** $x$ (a weight or an activation)
2: **Require:** scale $\alpha$, grids $\widehat{\mathcal{G}} = [g_0, \cdots, g_{2^b-1}]$
3: $\widehat{x} = \alpha \cdot \text{round}\left( \dfrac{x}{\alpha} \right)$
4: **Output:** $\min(\max(\widehat{x}, g_0), g_{2^b-1})$

---

