# OpenReview forum: "Semi-Relaxed Quantization with DropBits: Training Low-Bit Neural Networks via Bitwise Regularization"
_ICLR.cc/2021/Conference — Reject_

### Official Review · AnonReviewer3 · 2020-10-25
**Review of Paper3191**

**Rating:** 5
**Confidence:** 4

**Review:**

This paper deals with network quantization. It proposes Semi-Relaxed Quantization (SRQ) that uses a multi-class straight-through estimator to effectively reduce the bias and variance, along with a new regularization technique, DropBits that replaces dropout regularization to randomly drop the bits.  Extensive experiments are conducted to validate our method on various benchmark datasets and network architectures.

Pros:
- Reducing variance of gradient estimator in Gumbel of RQ using multi-class STE.
- A novel dropbits technique to reduce bias in gradient estimator of SRQ, as well as supporting the mixed-precision scheme.
- A "quantized winning tickets" is introduced to train probs of binary masks so that learning proper bitwise for each layer.

Cons:
- The SRQ process seems not to be novel enough. Actually here pi is just a multi-class sigmoid-output, and y is derived directly by argmax(prob_pi) in forward process (and calculate gradients using STE.). This
is quite similar to existing quantization methods using softmax + STE. [1]
- Only 3/4 bit results of Res18/MobileNetV2 showed in ImageNet. I'd appreciate it if authors could offer more quantitive analysis on more architectures and tasks.
- More comparisons on latency/energy/flops during the training and evaluation process should be provided to validate the SRQ + DropBits.

[1] Hardware-aware Softmax Approximation for Deep Neural Networks. Xue Geng et al.

***After rebuttal and discussion

 This paper proposes a new network quantization framework. In particular, the proposed DropBits is somewhat novel. However, it lacks sufficient and accurate analysis of SRQ+DropBits.  For example,  why SRQ can reduce quantization error has not been well motivated and explained.  The definition of distribution bias is still unclear.  I think that an accurate description of terminology is crucial and required for scientific research. Hence, the paper still needs minor polishing for publishing. I would like to decrease my rating to 5.

---

> ### Author Response · Authors · 2020-11-21
> **Dear Reviewer 3**
>
> Dear Reviewer 3,
>
> Thank you for your constructive feedback.
>
> ---------------------------------------------------------------------------------------------------------------------------
>
> [Q1: The SRQ process seems not to be novel enough. This is quite similar to existing quantization methods using softmax + STE. [1]]
>
> Response: Thank you for letting us know [1]. We added it to the reference in the revision.  However, our SRQ is quite different from [1]. The main purpose of [1] is to design a cost-efficient “softmax layer” for inference. Toward this, [1] introduces a hardware-aware softmax layer, which is achieved by reducing the operand bit-width and approximating the operations in softmax (such as exponential and division) via a look-up table. On the other hand, our work focus on network quantization of **all** layers in extremely low-bitwidth with the novel multi-class STE. Also, our experiments consider significantly lower bit-width than those of [1].
>
> ---------------------------------------------------------------------------------------------------------------------------
>
> [Q2: I'd appreciate it if authors could offer more quantitative analysis on more architectures and tasks.]
>
> Response: We agree with the reviewer’s opinion. However, particularly because our model as well as all major baselines such as RQ take much longer than the regular training process, it is quite difficult to show results beyond the experiments generally conducted in the existing literature. For large scale ImageNet experiments, the majority of papers in this field provided the result of *either* ResNet (deeper than ResNet-18) or  MobileNet. In addition to the result of MobileNet V2 we already provided in the original submission, now we are trying to quantize ResNet-34. But, due to the time constraint of the rebuttal period, we will add this result in the final revision.
>
> ---------------------------------------------------------------------------------------------------------------------------
>
> [Q3: More comparisons on latency/energy/flops during the training and evaluation process should be provided to validate the SRQ + DropBits.]
>
> Response: Note that counting FLOPs to compare two algorithms generally known to be meaningful *only* when both algorithms employ exactly the same operations. The *learning* process of RQ and SRQ not only consists of different sets of operations, but also includes several PyTorch built-in functions beyond the matrix/vector multiply-add operations, which can further mislead us. Instead, we compare the training time per epoch. For training ResNet-18 on ImageNet in 3-bit with a batch size of 128, the training time of SRQ + DropBits is around 90 minutes per epoch, which is 15% longer than that of RQ, about 76 minutes.
>
> Meanwhile, in the context of network quantization, latency/energy/flops at inference are much more important than those during the training phase since we use deployed models. In this sense, the inference procedure of SRQ + DropBits is the same as that of other uniform quantization methods, so are latency/energy/flops at inference.

---

### Official Review · AnonReviewer1 · 2020-10-26
**Interesting ideas, but needs some more work**

**Rating:** 6
**Confidence:** 4

**Review:**

### Summary
This work presents 1) Semi-Relaxed Quantization (SRQ), a method that targets learning low-bit neural networks, 2) DropBits, a method that performs dropout-like regularization on the bit width of the quantizers with an option to also automatically optimise the bit-width per layer according to the data, and 3) quantised lottery ticket hypothesis. SRQ is an extension of Relaxed Quantization (RQ), which is prior work, in two ways; firstly the authors replace the sampling from the concrete relaxation during training to deterministically selecting the mode (which is non-differentiable) and, secondly, they propose a specific straight-through gradient estimator (STE) than only propagates the gradient backwards for the elements that were selected in the forward pass. DropBits is motivated from the perspective of reducing the bias of the STE gradient estimator by randomly dropping grid points associated with a specific bit-width and then renormalising the SRQ distribution over the grid. This essentially induces stochasticity in the sampling distribution for the quantised value (which was removed before by selecting the mode in SRQ). The authors further extend DropBits in a way that allows for learning the drop probabilities for each bit-width, thus allowing for learning mixed-precision networks. Finally the authors postulate the quantised lottery ticket hypothesis, which refers to that “one can find the learned bit-width network which can perform better than the network with the same but fixed bit-widths from scratch”.

### Pros
- This work provides a set of additions that improves upon prior work
- The DropBits method is novel and allows for learning the bit-width in a straightforward manner
- The results improve upon recent works that learn quantised neural networks

### Cons
- Some claims from the authors are misleading while others are not precise
- The computational complexity of the method is not discussed
- Some experimental settings might not be consistent with some of the baselines

### Detailed feedback
This work tackles the problem of learning quantized neural networks and the authors show empirically that their proposed method achieves good results. The DropBits extension is particularly interesting in that it allows for learning the appropriate bit-width of a given tensor via traditional pruning approaches. I also like the fact that the authors explain illustratively their proposed approach via several figures, which provide a nice boost to clarity. Nevertheless, I believe that there are still some important aspects that need to be addressed before I recommend acceptance for this work.

First of all, the comparison against the prior work on figure 1 is misleading; the authors compare the *entire categorical distribution* (i.e., the one obtained after discretising the logistic onto the quantization grid) of SRQ at Fig 1(b) with a *single sample* from the concrete relaxation of the same distribution at Fig. 1(a) right for RQ. In fact, the underlying categorical distribution will be the same for both SRQ and RQ in the specific example of figure 1. Furthermore, it is worthwhile to notice that the underlying categorical distribution (i.e., pre relaxation) does have support for the value of -a (as the p(g_i = -a) is nonzero at Fig.1 (b)), thus it is not unreasonable that there are specific samples which lead to the quantised value being -a, thus incurring larger quantization loss.

Furthermore, I believe that the discussion about SRQ misses some important points that would improve the clarity of the work if they are addressed. Selecting the most probable point in the categorical distribution for the forward pass is equivalent to rounding to the nearest grid point, which can be done much more efficiently than computing the entire categorical over the grid and then taking the argmax. In addition, this is also the same as taking sigma -> 0 for the logistic distribution that is to be discretized in the forward pass. Finally the authors argue that their novel multi-class STE reduces the variance of the gradient estimator but no formal justification is given apart from some hand-wavy arguments. Why is the variance lower if the gradient only flows through to r_imax? Furthermore, for the second point with respect to the benefits of the multi-class STE; while it does seem desirable that it aggressively clusters the weights and activations around the grid-points, I wonder how much can that hinder convergence. Do you ever observe that the weights can be prematurely “stuck” (and thus lead to a bad local minimum) and do the weights ever move further away than just the closest grid point? DropBits could potentially help with the latter part, but it would be interesting to see what happens without it.

DropBits in my opinion is the main novel idea of this work, and it is an interesting way to learn the bit-precision of each tensor in the network. I have two main points for this section in general and DropBits in particular. The main motivation behind DropBits seems to be converting the sampling distribution of SRQ (which is deterministic) to a stochastic one. If this is the case, then why have it be deterministic in the first place? You could just sample from the original categorical distribution (by, e.g., using the Gumbel-Softmax STE which gives samples exactly on the grid) in the forward pass and use your multi-class STE approximation in the backward pass.  It would be interesting to see how that fares with SRQ + DropBits and would highlight whether the main benefit of DropBits was the regularization aspect (and not that of improving the sampling distribution). As for DropBits in particular; it seems that you drop bits independently with each of the gates z_1, z_2, … (i.e., figure 3). If this is the case, then you could end up with a non-uniform grid that cannot be exactly represented as a fixed point tensor (e.g., on figure 3b you could have z_1 = 0 and z_2 = 1). If this is the case, then comparison against other approaches that use uniform quantization is not apples-to-apples.

Finally, a couple of other things that I believe should be addressed; the authors don’t make any discussions about the computational and memory complexity of the resulting algorithm. It seems that for every individual weight and activation they first have to construct the categorical distribution over the entire grid (which can quickly become very large, e.g., for 8 bits there are 256 categories), in order to take the weighted sum. This doesn’t seem to scale very well. How expensive is something like this in practice and how long do experiments take on, e.g., Imagenet? Furthermore, the quantised lottery ticket hypothesis (QLTH) is a bit peculiar. The original lottery ticket hypothesis (LTH) was about finding sparse networks at initialisation that can be trained from scratch and achieve the same accuracy as the original dense equivalents. This is different than what the authors articulate here, specifically that it is about finding sparse networks that are easier to train compared to sparse networks obtained from pruning. As a result, their QLTH seems to state the opposite than what the original LTH was about; it states that a QLT is obtained when you manage to find a network X that a.) has smaller bit width than the original network and b.) has better performance than a network initialised to the bit-width of X and trained to convergence. Following the arguments of the LTH, I would expect a QLT to be obtained when you can quantise a neural network to a specific bit-width at initialisation and when you train from scratch that particular quantised network, you obtain the same performance as the full precision equivalent. I would thus encourage the authors to clarify this point and better align with the original LTH.

Based on the aforementioned points, I cannot at the moment recommend acceptance for this work. Nevertheless, as I believe DropBits is an interesting idea, I would encourage the authors to put in the effort and rework the paper by addressing these points over the rebuttal.

---

> ### Author Response · Authors · 2020-11-21
> **Dear Reviewer 1 [1]**
>
> Dear Reviewer 1,
>
> Thank you for your constructive feedback.
>
> [Q1: The comparison against the prior work on figure 1 is misleading.]
>
> Response: As the reviewer astutely pointed out, we only consider a single sample in Figure 1 in the original submission. In the revision, we exhibit more cases for RQ so as not to be misunderstood.
>
> Also as mentioned by the reviewer, since the underlying categorical distribution has support for $-\alpha$ and $-2\alpha$, it appears reasonable that there are specific samples which lead to the quantized value being below zero as shown in Figure 1-(b,d,e) in the revision. It may be okay *on average*, but RQ computes *only one* sample in each forward pass due to computational burden, which can accidentally lead to very large quantization error for this particular sample.
>
> On the other hand, SRQ deliberately performs biased estimation on the underlying categorical distribution to prevent large quantization error from even occurring in the forward pass while sharing this underlying distribution with RQ in the backward pass. Instead, we devised DropBits to reduce the incurred bias of SRQ. For more details, see new Section 3.4 in the revision.
>
> ----------------------------------------------------
>
> [Q2: Selecting the most probable point in the categorical distribution for the forward pass is equivalent to rounding to the nearest grid point.]
>
> Response: As the reviewer mentioned, the forward pass of SRQ is equivalent to that of rounding to the nearest grid point, but the backward pass of SRQ is totally different from that of rounding. As mentioned in the last paragraph of Section 3.2, the backward pass of SRQ enables weights and activations to be cohesively clustered around grid points, whereas there is no guarantee that rounding to the nearest grid point could cluster them around grid points. Such a difference in the backward pass makes SRQ perform well.
>
> As AC and Reviewer 1 mentioned, the forward pass of SRQ is identical to a deterministic quantization like rounding to the nearest bin. It may seem a bit complicated for a forward pass, but expressing the nearest grid selection in this equivalent form (the combination of logistic and argmax) is essential to derive our backward pass. As a result, the backward pass of SRQ is totally different from that of other deterministic quantization methods. As mentioned in the last paragraph of Section 3.2, the backward pass of SRQ enables weights and activations to be cohesively clustered around grid points, whereas there is no guarantee that other deterministic quantization methods could cluster them around grid points.
>
> In order to improve clarity by giving high-level ideas for forward and backward passes of our model in advance, we slightly changed the opening of Section 3 as
> “propose Semi-Relaxed Quantization, which selects the nearest grid point in the forward pass to decrease the quantization error. To make it learnable and to cluster compressed parameters cohesively, SRQ expresses the nearest grid selection of the forward pass as the equivalent form, the combination of logistic distribution and argmax, and performs the backward pass on it.”
>
> --------------------------------------------------
>
> [Q3: In addition, this is also the same as taking sigma -> 0 for the logistic distribution that is to be discretized in the forward pass.]
>
> Response: As the reviewer mentioned, the result of taking $\sigma = 0$ is the same as that of taking argmax. However, exact zero $\sigma$ makes the network untrainable. To be concrete, the probability distribution of $\widetilde{x}$ for $\sigma = 0$ becomes the Dirac delta function as $p(\tilde{x}) = \delta(x)$ whose CDF is just the step function. Then, the gradient ${\partial \pi_{i_{max}} \over \partial x}$ becomes exactly zero almost everywhere, so is the gradient ${\partial \mathcal{L} \over \partial x}$. Thus, the statement is absolutely correct but the network parameters would never be updated via gradient-based optimization.
>
> ---------------------------------------------------------------
>
> [Q4: Why is the variance lower if the gradient only flows through to r_imax?]
>
> Response: As we mentioned in our response to AC, the variance of gradient estimator using our multi-class STE is exactly zero due to the fact that there is no randomness in SRQ. Hence, the variance of our gradient estimator is trivially lower (actually, exactly zero) than that of RQ based on sampling from Gumbel distribution. We explained it in more detail in the second last paragraph of Section 3.2 in the revision.
>
> -----------------------------------------------------------

---

> > ### Author Response · Authors · 2020-11-21
> > **Dear Reviewer 1 [2]**
> >
> > [Q5: I wonder how much can that hinder convergence. Do you ever observe that the weights can be prematurely “stuck” (and thus lead to a bad local minimum) and do the weights ever move further away than just the closest grid point?]
> >
> > Response: In case of large learning rates, we have indeed observed that SRQ seemed to be stuck in a bad local optimum by moving network parameters further away than the nearest grid point. To confirm this phenomenon, we choose a random coordinate of LeNet-5 and see where this coordinate is quantized just after a single training iteration on MNIST dataset. With a learning rate of 1e-2, this weight parameter that should be quantized to $-2\alpha$ is quantized to $-\alpha$ after an update, which means that a weight can be moved further away than the closest grid point, thus being able to be stuck in a bad local minimum.
> >
> > -----------------------------------------------------
> >
> > [Q6: Why have it to be deterministic in the first place? You could just sample from the original categorical distribution (by, e.g., using the Gumbel-Softmax STE which gives samples exactly on the grid) in the forward pass and use your multi-class STE approximation in the backward pass.]
> >
> > Response: As described in Section 3.4 in the revision, our SRQ + DropBits shows smaller quantization error, variance of gradients, and bias than RQ, while maintaining stochasticity. For this reason, we employ the deterministic scheme in the first place and then encourage stochasticity via DropBits.
> >
> > To show the effectiveness of SRQ + DropBits further, we empirically compare it with the case when the Gumbel-Softmax STE is used in the first place instead of DropBits (this is the setting the reviewer mentioned), denoted as “Gumbel-Softmax + multi-class STE” in the table below.
> >
> > |       | # Bits W./A. | Gumbel-Softmax + multi-class STE | SRQ + DropBits |
> > |-------|--------------|----------------------------------|----------------|
> > |       | 4/4          | 0.57                             | 0.53           |
> > | MNIST | 3/3          | 0.65                             | 0.58           |
> > |       | 2/2          | 0.74                             | 0.63           |
> >
> > |       | # Bits W./A. | Gumbel-Softmax + multi-class STE | SRQ + DropBits |
> > |-------|--------------|----------------------------------|----------------|
> > |          | 4/4          | 8.25                             | 6.85           |
> > | CIFAR-10 | 3/3          | 8.44                             | 6.94           |
> > |          | 2/2          | 10.41                            | 7.51           |
> >
> >
> > Although employing our multi-class STE in the backward pass, Gumbel-Softmax + multi-class STE performs worse than SRQ + DropBits. This is primarily due to the fact that Gumbel-Softmax STE still incurs a large quantization error like RQ. We included this result in Appendix D in the revision.
> >
> > ---------------------------------------------------
> >
> > [Q7: When z_1 = 0 and z_2 = 1 on figure 3b, you could end up with a non-uniform grid that cannot be exactly represented as a fixed point tensor]
> >
> > Response: At inference, $(Z_1, Z_2) = (0, 1)$ might occur as you pointed out. In this case, for a 3-bit quantization example, only $[g_0, g_1, g_3, …, g_7]$ (the grid point $g_2$ is excluded) would be considered in the testing phase. Nevertheless, this still can be thought of as a kind of uniform quantization where no weight is quantized to $g_2$ in its realization. Even though such a case occurs, we can just treat our method as a uniform quantization method in evaluation. We believe that it is not unfair to compare SRQ + DropBits with other uniform quantization baselines since SRQ + DropBits finally uses fewer number of grid points than those approaches.
> >
> > --------------------------------------------------
> >
> > [Q8: How expensive is something like this in practice and how long do experiments take on, e.g., Imagenet?]
> >
> > Response: The time spent on our ImageNet experiments greatly varies depending on the number and type of GPUs used, GPU parallelism, and so on. In our setting, when training ResNet-18 on ImageNet in 3-bit with a batch size of 128, the training time of SRQ + DropBits is around 90 minutes per epoch, which is 15% longer than that of RQ, about 76 minutes.
> >
> > As AC mentioned, considering two neighboring bins could be one of possible designs for our SRQ framework so as to improve the computation and memory complexity, but we consider a categorical distribution over all grid points for DropBits and heterogeneous quantization in Section 3.3 and 3.4.
> >
> > --------------------------------------------------------
> >
> > [Q9: Their QLTH seems to state the opposite than what the original LTH was about.]
> >
> > Response: Thank you for the helpful suggestion. The definition of our hypothesis for quantization is somewhat different (in some sense opposite, as the reviewer said) from that of the original one for pruning. To avoid the confusion, we remove the name ‘lottery ticket’ from our hypothesis in the revision.

---

> > > ### Comment · AnonReviewer1 · 2020-11-23
> > > **Response to rebuttal**
> > >
> > > Thank you for addressing my comments, the submission is in better shape now. The answers to Q1, Q2 are clear enough and for Q3, I was suggesting of using a sigma -> 0 in the forward pass only (and using a sigma different than zero in the backward pass). Explanations and additions to the submission according to Q4, Q5, Q8 are good and the extra experiments on Q6 are appreciated.
> > >
> > > My main concern is still about Q7 and Q9.
> > >
> > > Q7: This might not be as simple; for weights, this can be the case, but then you need to report in the results the bit width that includes these ‘dummy’ grid points, as that would be the format of the tensor in hardware. For activations this becomes trickier as for what you are saying to hold, you need to have activations that are not near g2, as when you round to nearest in hardware you would incorrectly assign g2 instead of, e.g.,  g3. It would thus be interesting to check performance of SRQ + dropbits, when you manually select the values of the gates z , such that the resulting tensor is a valid fixed point tensor. If that doesn’t decrease performance, then this might not be an issue in practice (i.e., when deploying on a fixed point device) and comparisons will be valid.
> > >
> > >
> > > Q9: The changes make this claim more clear, although you still mention in various places “quantized winning tickets” so there can still be confusion.
> > >
> > > Overall, there have been quite a few updates on the submission (some of which could be called major), but it has improved compared to its previous state. While I cannot fully recommend acceptance yet due to Q7, I will increase my score to a 6.

---

> > > > ### Author Response · Authors · 2020-11-25
> > > > **Dear Reviewer 1 [3]**
> > > >
> > > > Dear Reviewer 1,
> > > >
> > > > Thank you for your detailed response.
> > > >
> > > > [Q7. When z_1 = 0 and z_2 = 1 on figure 3b, you could end up with a non-uniform grid that cannot be exactly represented as a fixed point tensor.]
> > > >
> > > > Thank you for the clarification; we now exactly understand what you are concerned about. All the values we reported in the original submission are already the results in a situation where all grid points are allowed to be selected in the inference time (regardless of the learned grid points with DropBits). Hence, there are no issues in fairness.
> > > >
> > > > Moreover, we in fact considered DropBits **only for network weights** throughout all our experiments. This is primarily due to the fact that quantization grids for activations start from zero whereas those for weights are symmetric with respect to zero. As DropBits is symmetrically designed, we simply applied it only for weights. We are sorry that we did not mention it explicitly. We added it in the last paragraph of Section 3.3.
> > > >
> > > > -----------------------------------------------------------------------------------------------------------------------
> > > >
> > > > [Q9. Their QLTH seems to state the opposite than what the original LTH was about.]
> > > >
> > > > Thank you for checking this issue carefully. As your suggestion, we replaced the word “quantized winning tickets” with “quantized sub-networks” in the revision for more clarity. We will continue to proofread regarding on this issue.

---

### Official Review · AnonReviewer4 · 2020-10-28
**Well written and organized paper**

**Rating:** 7
**Confidence:** 2

**Review:**

This paper proposed a novel network quantization method to reduce the bit-lengths of the network weights
and activations, which is one of the most important problem for resource-limited devices.
The presentation of this paper is well written and organized.
The problem definition is clear, i.e., relaxed quantization with the Gumbel-Softmax relaxation suffers from bias-variance trade-off depending on the temperature parameter of Gumbel-Softmax.
The proposed method to solve this problem is reasonable.
The experimental results are also convincing.

Minor comments:
1. In Table 3, "MNIST" should be LeNet-5.
2. Section 3.4 is a little dense.
3. I understand that there is little space, but embedding formulas in sentences is difficult  to read.
4. I recommend you to show the result only using SQR for  ImageNet  in Table 2 as in MNIST and CIFER10 in Table 1.

---

> ### Author Response · Authors · 2020-11-21
> **Dear Reviewer 4**
>
> Dear Reviewer 4,
>
> Thank you for your constructive feedback.
>
> We are sorry for the inconvenience with typo and small embedding formulas. We corrected them in the revision.
>
> As your suggestion, we are conducting experiments on ImageNet by only using SRQ in 3-bit. When quantizing ResNet-18 to 3-bit by only using SRQ, top-1 error is 37.45% and top-5 error is 15.47% after performing fine-tuning throughout 110 epochs with a batch size of 128. Although SRQ in 3-bit already outperforms RQ in 4-bit, it seems that its performance could be enhanced as the training progresses further.
>
> Due to the time constraint of the rebuttal period and the fact that more epochs are required to train SRQ than SRQ + DropBits, we unfortunately have not yet completed this experiment, but we will add it in the final revision.

---

### Comment · Area_Chair1 · 2020-11-11
**Technical concerns about primary contribution**

I would like to bring into the attention of authors and reviewers the following technical concerns about the main part of the submission, which is the proposed "novel SRQ method that uses multi-class straight-through estimator to efficiently reduce the bias and variance".

The forward pass of SRQ (3) appears identical to a deterministic quantization (since the logistic density is symmetric and decreases monotonously away from the mode, argmax would be found in the nearest bin to the input x). This is already remarked by R1. If this is correct, the presentation through the categorical argmax is redundant and occludes clarity.

The choice of the backward pass of the estimator is not based on any formal reasoning. Sufficiently many unjustified heuristics exist already, e.g. the "ST Gumbel-Softmax Estimator" (Jang, 2018) samples discrete values on the forward pass, empirically mitigating the problem that the relaxed samples from the continuous Gumbel-softmax are real-valued. A justification is needed why the proposed estimator would be any better? This is promised in the abstract but the paper fails to show it (see below).

When expanding the gradient estimate in the end of sec. 3.2 the authors show that the gradient vanishes when x is close to a quantization point. This means when the pre-activation happens to be discrete, the gradient is zero, and therefore the network does not learn (through this pre-activation). It is not clear what is achieved by having this property, surely not a gradient-based learning in a common sense. From a reasonable estimator for quantization, I would expect at least that in the limit of a very dense quantization (large number of bits), it would coincide with the continuous gradient. However this estimator approaches zero in this limit (when sigma is fixed). What does it do in the case sigma is tied to the bin size? If it is considered for a small number of bits only, a formal reasoning and properties are needed.

When referring to the variance and bias of an estimator, I expect a clear definitions of what is assumed to be random. Since both the forward and the backward passes are deterministic (nothing sampled), there is nothing random. The variance is not just "low" as described in the submission, it is zero. When speaking of the bias of an estimator I expect to see a definition of the reference for the bias, i.e.  the true gradient of some differentiable function, with respect to which the bias can be measured. Nothing like this is defined. Therefore the claims of reducing variance and bias in the abstract are not substantiated and misleading.

Some minor comments:
It is not clear why to renormalize the bin probabilities (sec. 3.2). Would it not be better to take the tails of the distribution on the ends of the quantization interval, i.e. \pi_0 = F[0] and pi_{2^b-1} = 1 - F[2^b-1]?
The term "large quantization loss" is unclear, what does it mean?
When sigma is tied to the bin size as sigma = alpha/3, the support of the noise distribution can be cut to say +- 3 sigma = +1 one bin. In this case, there is no need to consider a categorical distribution on all bins, just the neighboring bins. I suppose that can improve computation complexity of the method. Maybe considering just two neighboring bins and a binary distribution on them would do the job while being much more clear?

Since these points critically affect the major contribution of the paper (the new multi-class straight-through estimator). I believe a major revision would be necessary for this part alone. I propose to reject.

That said, I invite the authors and reviewers to discuss all aspects of the submission. I will rejoin in the next phase.

---

> ### Author Response · Authors · 2020-11-21
> **Dear Area Chair [1]**
>
> Dear Area Chair,
>
> Thank you for your constructive feedback.
>
> [Q1: The forward pass of SRQ (3) appears identical to a deterministic quantization and if this is the case, the presentation is redundant and occludes clarity.]
>
> Response: As AC and Reviewer 1 mentioned, the forward pass of SRQ is identical to a deterministic quantization like rounding to the nearest bin. It may seem a bit complicated for a forward pass, but expressing the nearest grid selection in this equivalent form (the combination of logistic and argmax) is essential to derive our backward pass and we do not agree that this rewrite is redundant. As a result, the backward pass of SRQ is totally different from that of other deterministic quantization methods. As mentioned in the last paragraph of Section 3.2, the backward pass of SRQ enables weights and activations to be cohesively clustered around grid points, whereas there is no guarantee that other deterministic quantization methods could cluster them around grid points. As stated below, even if $x$ is close to a quantization point, $\alpha$ can be trained so that $x$ can be trained again after $\alpha$ is updated. Such a difference in the backward pass makes SRQ perform well.
>
> In order to improve clarity by giving high-level ideas for forward and backward passes of our model in advance, we slightly changed the opening of Section 3 as
> “propose Semi-Relaxed Quantization, which selects the nearest grid point in the forward pass to decrease the quantization error. To make it learnable and to cluster compressed parameters cohesively, SRQ expresses the nearest grid selection of the forward pass as the equivalent form, the combination of logistic distribution and argmax, and performs the backward pass on it.”
>
> ---------------------------------------------------------------
>
> [Q2: The zero gradient near grid points makes the network untrainable which would be not a gradient-based learning. What is achieved by having this property?]
>
> Response: When $x$ is close to a quantization point, ${\partial \mathcal{L} \over \partial x}$ is almost zero, but $\alpha$ is still trained. Therefore, if $\alpha$ is updated, there is a gap between $x$ and $\alpha$ so that $x$ can be trained. Hence, the network will continue to train until it finds the optimal $\alpha$. We included this fact at the end of Section 3.2, which we believe is a marginal revision.
>
> -------------------------------------------------------------
>
> [Q3: From a reasonable estimator for quantization, I would expect at least that in the limit of a very dense quantization (large number of bits), it would coincide with the continuous gradient. However this estimator approaches zero in this limit (when sigma is fixed). What does it do in the case sigma is tied to the bin size?]
>
> Response: In a situation where we directly learn $x$ that can reduce the loss $L$, we agree that it is natural to coincide the gradient of $x$ with the continuous version when the number of grids goes to the limit. However, as can be seen from the analysis of the gradient, our quantization algorithm is a method of adjusting the grid so that when $x$ are clustered into the grid, the resulting network has small loss $L$. In fact, this clustering effect is one of the big advantages of our algorithm. When the number of bits goes to infinity, all real values become grid points themselves so that $x$ is already clustered around grid points. In such a case, it is natural that our ``clustering-based’’ quantization is not working.
>
> -----------------------------------------------------------
>
> [Q4: The variance is not just "low" as described in the submission, it is zero.]
>
> Response: In SRQ, the variance of gradients is indeed zero as you mentioned. We are sorry for the confusion. We corrected accordingly in the revision.
>
> ----------------------------------------------------------
> [Q5: When speaking of the bias of an estimator I expect to see a definition of the reference for the bias.]
>
> Response: We are sorry to define bias implicitly. Strictly speaking, we somehow abuse the notation in the original version. In section 3.2, the bias actually means “quantization error”, the difference between an input value and its quantized value. In section 3.3, however, the bias means the difference between the original categorical distribution and the distribution approximated by an algorithm such as RQ, SRQ, and SRQ + DropBits, $\mathbb{E}[p] - \pi_{origin}$ where the vector $\pi_{origin} := (\frac{\pi_i}{\sum_{i=0}^{2^b-1}\pi_j})_{i=0}^{2^b-1}$ is the original categorical distribution and the vector $p := (p_i)_{i=0}^{2^b-1}$ is the approximated one by a certain algorithm. We will only use the term bias to indicate this. Accordingly, we replaced the term bias with quantization error in section 3.2.
>
> ---------------------------------------------------------

---

> > ### Author Response · Authors · 2020-11-21
> > **Dear Area Chair [2]**
> >
> > [Q6: The choice of the backward pass of the estimator is not based on any formal reasoning. A justification is needed why the proposed estimator would be any better.]
> >
> > Response: Thank you for your comment. Although SRQ + DropBits shows smaller quantization error, bias, and variance than RQ, we did not express it explicitly in the original submission. We added the new Section 3.4 on it in the revision.
> >
> > To summarize, although RQ is a quantization method based on a probabilistic approach, it suffers from large quantization error as we highlighted in Section 3.2. To resolve this issue, we introduce a novel multi-class straight-through estimator with a deterministic forward procedure. Our STE design has several advantages over RQ baseline. First, the variance of gradient estimator for SRQ (from backward pass in Eq. (3) in our paper) is exactly zero since there is no randomness whereas RQ must sample from Gumbel random variables.
> > Moreover, SRQ quantizes a parameter to its nearest grid point, so the quantization error of SRQ is always smaller than that of RQ.
> >
> > However, SRQ is biased to the mode by construction. As a remedy for this issue, we propose DropBits. Remarkably, SRQ + DropBits shows smaller quantization error, bias, and variance than RQ as shown in Figure 6 in the revision (see Section 3.4 for details).
> >
> > ---------------------------------------------------------------------------------------------------------------------------
> >
> > [Q7: Would it not be better to take the tails of the distribution on the ends of the quantization interval, i.e. \pi_0 = F[0] and pi_{2^b-1} = 1 - F[2^b-1]?]
> >
> > Response: Our SRQ framework models the probability of being quantized to each grid point through the Logistic distribution which is “symmetric” around the mean. In Figure 1, if \pi_0 = F[0] and pi_{2^b-1} = 1 - F[2^b-1], then $[\pi_0, \pi_1, \pi_2, \pi_3] = [0.0025, 0.0450, 0.4526, 0.5000]$. For a weight ✭, whose value is ${\alpha \over 2}$, the distance between ✭ and 0 is the same as the distance between ✭ and $\alpha$, but $\pi_2$ is not equal to $\pi_3$, which contradicts to our design using Logistic distribution. Thus, we did not consider such a case.
> >
> > ---------------------------------------------------------------------------------------------------------------------------
> >
> > [Q8: The term "large quantization loss" is unclear, what does it mean?]
> >
> > Response: We are sorry for the confusion. It means “large quantization error”, the difference between an input value and its quantized value, $|\widehat{x} - x|$. We replaced it with “large quantization error” in the revision.
> >
> > ---------------------------------------------------------------------------------------------------------------------------
> >
> > [Q9: Maybe considering just two neighboring bins and a binary distribution on them would do the job while being much more clear?]
> >
> > Response: As your suggestion, only considering two neighboring bins could be one of possible designs for our SRQ framework, which could improve the computation and memory complexity. But, for DropBits and heterogeneous quantization in Section 3.3 and 3.4, we need to consider a categorical distribution over all bins.

---

> > > ### Comment · Area_Chair1 · 2020-11-23
> > > **Acknowledging Response**
> > >
> > > I thank the authors for their response. I appreciate the update of the paper with the intuition behind the method made somewhat more clear, but I am afraid it still does not approach the clarity and the correctness necessary for publishing.
> > >
> > > The Gumbel-Softmax (GS) estimator (Jang et al. 2017) estimates the gradient of the expectation of the loss, where the expectation in the case of the submission is over the logistic noises added prior to quantizing. Given that,
> > > - The quantization error defined in the new 4.3, is independent of the way the gradient is estimated. It is defined by the stochastic relaxation alone. Therefore the "quantization error of GS estimator" is thus undefined and cannot be decreased, contrary to what the abstract claims. Note also that the normal operation mode for stochastic relaxation methods is to start with a higher uncertainty and during the optimization to converge to more deterministic models (implying lower quantization loss). To make it converge faster to more deterministic distributions, prior work sometimes adds entropy as a regularizer (e.g. Shayer et al. 2018 cited in the submission). Minimizing entropy leads to more deterministic models.
> > > - The bias is already defined by Jang et al. to be the difference of the expected gradient by the estimator and the true gradient, which can be obtained using an unbiased estimator with sufficiently many samples.
> > > - The submission introduces a new notion of bias, let us term it a "distribution bias", that also has nothing to do with the gradient. Introducing this notion is not clearly justified.
> > > - In total, instead of bias and variance that was in the consideration before (initial submission and prior art), we have now 4 different quantities, two of which are not properties of gradient estimators. The comparison in 4.3 in the revised version is somewhat interesting, but entangling the problem more instead of clarifying it. The notion of estimator appears to be significantly misused to denote some algorithm for forward and backward pass with some desirable properties for different purposes.
> > > To reduce the quantization error of the model, the only thing that can be done is to reduce the variance of logistic noises $\sigma$. And indeed if $\sigma$ approaches zero, and say we introduce an "estimator" that samples from the true categorical distribution on the forward pass and returns zero gradient on the backward pass, it has minimal quantization error, zero distribution bias as measured in section 4.3 and zero variance. Clearly, this method does not resolve the bias-variance trade-off of Gumbel-Softmax.
> > >
> > > Some further aspects not addressed:
> > > - Straight-Through Gumbel-Softmax estimator (Jang et al. 2017, sec. 2.2) not discussed / compared to
> > > - Q2 (zero gradient). The answer does not resolve the issue that the gradient in $x$, as estimated, is zero, which appears to be wrong regardless the gradient in $\alpha$.
> > >
> > > Clarification:
> > >
> > > - Q7: Would it not be better to take the tails of the distribution on the ends of the quantization interval, i.e. \pi_0 = F[0] and pi_{2^b-1} = 1 - F[2^b-1]?
> > >
> > > What I meant here is to define quantization intervals as
> > > $(-\infty, g_0], [g_0, g_1], \dots [g_{2^b-1}, \infty)$,
> > > then the probability to fall inside end intervals will be given similarly to (1) by differences of noise cdf. Then the renormalization of distribution will not be needed, since the intervals form a partition of the real line. This is not a major issue, just suggestion.

---

> > > > ### Author Response · Authors · 2020-11-24
> > > > **Dear Area Chair [3]**
> > > >
> > > > Thank you for your constructive response.
> > > >
> > > > [Q. The answer does not resolve the issue that the gradient in $x$, as estimated, is zero, which appears to be wrong regardless of the gradient in $\alpha$.]
> > > >
> > > > We would like to improve our paper according to your comments. However, due to the impending deadline, we first ask you this question. We believe that the explanation in Q2 is convincing enough to resolve the issue when $x$ is close to a quantization grid point and nothing is wrong. Could you tell us what seems to be wrong in detail?
> > > >
> > > > We will respond to your remaining questions as soon as possible.

---

> > > > > ### Author Response · Authors · 2020-11-25
> > > > > **Dear Area Chair [4]**
> > > > >
> > > > > Dear Area Chair,
> > > > >
> > > > > Thank you for your constructive response.
> > > > >
> > > > > - We apologize for the overclaim that we have resolved the bias-variance tradeoff of Gumbel-Softmax estimator in the abstract. We accordingly revised those parts so that now we have “our method can be better than other methods in terms of bias in a distribution aspect”. We also replaced “bias” with “distribution bias”, as you suggested.
> > > > >
> > > > > - However, we want to emphasize that not only GS-ST mentioned by AC, but also GS estimator is a **biased** estimator (see ❉ below) and in that sense, it is also heuristic and not mathematically justified. Our paper also does not focus on mathematically proving that our estimator has better bias-variance tradeoff than GS or GS-ST. Instead, we indirectly support the fact that our method performs better in quantization by experimentally demonstrating that under certain conditions, our method has a better quantization error than RQ but possibly with a comparable bias/variance tradeoff (see Q3 for detailed issue on comparing bias/variance tradeoff across methods).
> > > > >
> > > > > ❉  The gradients of Concrete relaxations are biased with respect to the original discrete objective, but they are low variance unbiased estimators of a continuous surrogate objective. (Section 6 of Maddison et al., 2017)
> > > > >
> > > > > ----------------------------------------------
> > > > >
> > > > > [Q1. The quantization error defined in the new 3.4, is independent of the way the gradient is estimated. Therefore, the "quantization error of GS estimator" is thus undefined and cannot be decreased, contrary to what the abstract claims. The notion of estimator appears to be significantly misused to denote some algorithm]
> > > > >
> > > > > Note that the quantization error is defined for RQ using GS-estimator, not GS-estimator itself. We believe we have never defined and used the quantization error of GS-estimator, but we proofread it again to be more clear so that there is no misunderstanding.
> > > > >
> > > > > In addition, strictly speaking, GS estimator indicates the gradient estimation as mentioned by AC, but it is generally used in the broader meaning of methods or algorithms causing such gradient estimation. In fact, the authors of GS estimator broadly define it as the "procedure" of replacing non-differentiable categorical samples with a differentiable approximation, i.e., differentiable Gumbel-Softmax samples (the first paragraph in Section 2.1 of Jang et al., 2017). Hence in our context, the quantization error can be implicitly but clearly defined.
> > > > >
> > > > > ---------------------------------------------
> > > > >
> > > > > [Q2. The normal operation mode for stochastic relaxation methods is to start with a higher uncertainty and during the optimization to converge to more deterministic models.]
> > > > >
> > > > > We are aware that annealing the temperature $\tau$ of Gumbel-Softmax is widely employed as one of the methods to converge to a more deterministic distribution as the training progresses further. In Table 1 and Figure 7 of our paper, we already compared RQ+annealing with SRQ and SRQ + DropBits. As shown in Table 1, annealing $\tau$ brings about performance degradation of RQ, thereby showing worse performance than SRQ + DropBits as well as SRQ in most cases. As mentioned by AC, it may be possible to reduce the quantization error partially, but it does not make a big difference overall.
> > > > >
> > > > > ---------------------------------------------
> > > > >
> > > > > [Q3. The bias is already defined by Jang et al. to be the difference of the expected gradient by the estimator and the true gradient, which can be obtained using an unbiased estimator with sufficiently many samples.]
> > > > >
> > > > > In general, when the loss involves discrete random variables, the true gradient, the gradient of the expectation of the loss with respect to the parameter of discrete random variables, can be obtained by using existing stochastic gradient estimation techniques such as the score function estimator (see Equation (4) and (5) in Maddison et al., 2017). However, the reason why we did not explicitly compare the bias of gradient in the original submission was that distribution parameters (i.e. $\pi = \(\pi_i\)_{i=0}^{2^b-1}$) in our setting are NOT independent of network parameters $x$. In fact, $\pi$ is a function of $x$ in Eq. (1) in our paper. Given that an inverse of a function $\pi$ does not exist because $\pi_i$ is not one-to-one with respect to $x$ for each $i$, it is **not possible** to directly apply existing techniques such as the score function to compute unbiased estimators. In the new revision, we instead compute the bias of gradient with respect to $x$ as a proxy of that of w.r.t. $\pi$. As shown in Figure 8(b) in the revision, SRQ + DropBits and RQ exhibit comparable levels of bias with respect to $x$.

---

> > > > > > ### Author Response · Authors · 2020-11-25
> > > > > > **Dear Area Chair [5]**
> > > > > >
> > > > > > [Q4. A new notion of bias, "distribution bias", has nothing to do with the gradient. Introducing this notion is not clearly justified.]
> > > > > >
> > > > > > Our new notion of distribution bias is very loosely coupled with the gradient in the sense that both converge to zero if there is no approximation for the non-differentiable parts. However, it can be used to measure how much biased distribution is used in computing the gradient estimator. We admit that this is an indirect indicator that does not perfectly determine the superiority of the gradient estimators. We revised our paper so that this part is explicitly revealed.
> > > > > >
> > > > > > -----------------------------------------------------------------------------------------------------------------------
> > > > > >
> > > > > > [Q5. To reduce the quantization error of the model, the only thing that can be done is to reduce the variance of logistic noises $\sigma$. And indeed if $\sigma$ approaches zero, and says we introduce an "estimator" that samples from the true categorical distribution on the forward pass and returns zero gradient on the backward pass, it has minimal quantization error, zero distribution bias as measured in section 3.4 and zero variance.]
> > > > > >
> > > > > > Note that the learnable $\sigma$ is a parameter that determines the discrete distribution of the objective whose gradient we need to find, not just the parameter that determines our gradient estimate. If $\sigma$ is trained toward a very small value, it means that the discrete distribution **in our objective** is easy and mild to handle, and hence our estimator computing the gradient of objective with this "mild" discrete distribution can achieve small quantization error, distribution bias, and variance as AC mentioned. At the most extreme case $\sigma \rightarrow 0$, the Logistic distribution inside the objective boils down to Dirac delta function at the mode, so does the categorical distribution with $\pi_i$, hence its gradient can be easily computed. We do not believe there are any issues here.
> > > > > >
> > > > > > The problem is when $\sigma$ is large, that is, it is difficult to predict the gradient of objective including discrete distribution. Our goal is to devise an algorithm with small quantization error and small distribution bias even in this case and show high experimental performance.
> > > > > >
> > > > > > -----------------------------------------------------------------------------------------------------------------------
> > > > > >
> > > > > > [Q6. Straight-Through Gumbel-Softmax estimator (Jang et al. 2017, sec. 2.2) not discussed / compared to]
> > > > > >
> > > > > > The results of using Straight-Through Gumbel-Softmax estimators in quantization, named “RQ ST”, are also reported in Louizos et al. 2019. Originally, we did not include them because RQ with ST GS estimator did not make a meaningful difference. We added the results of RQ ST in Table 2 and Table 4 in the revision.

---

### Author Response · Authors · 2020-11-21
**Revision summary**

Thank you to all reviewers for their constructive and helpful feedback. Based on their comments, we revised the paper by making the following changes. The modified part is highlighted in blue in the revision.

Major updates:
- We have included the quantitative analysis on quantization error, bias, and variance of our method against RQ baseline (AC).
- We have replaced the “bias” and “quantization loss” with “quantization error” in abstract, Introduction, and Section 3.2 due to the ambiguity of the word “bias”. (AC)
- We have corrected the expression “low/small variance” to “zero variance” in Section 3.2. (AC)
- We have included the discussions on the learnability of our gradient estimator when the input $x$ is close to $g_{i_{max}}$ at the end of Section 3.2. (AC)
- We have changed Figure 1 and its descriptions by adding more examples of RQ according to different Gumbel sampling. (R1)
- We have explained the reason why the gradient variance of SRQ is zero in more detail. (R1)
- We have added the comparison of SRQ + DropBits with Gumbel-Softmax + multi-class STE in Appendix D. (R1)
- We have removed “lottery ticket” from our hypothesis in Section 3.6. (R1)
- We have included the reference “Hardware-aware Softmax Approximation for Deep Neural Networks. Xue Geng et al.” in Section 2. (R3)
- We have rearranged equations in Section 3.2 (Semi-Relaxed Quantization - Fixing Pitfalls of RQ) and Section 3.5 (Learning bit-width towards resource-efficiency) for readability. (R4)

We believe that our paper gets much stronger and clearer with this revision, thanks to the reviewers for constructive suggestions.

---

### Author Response · Authors · 2020-11-23
**The end of the discussion phase approaching**

Dear Reviewers and Area Chair,

Could you please go over our responses and the revision since we can have interactions with you only by this Tuesday (24th)? We have responded to your comments and faithfully reflected them in the revision, and provided additional experimental results. We sincerely thank you for your time and efforts in reviewing our paper, and your insightful and constructive comments.

Thanks, Authors

---

### Decision · Program_Chairs · 2021-01-07
**Final Decision**

**Decision:**

Reject

**Comment:**

# Summary
The paper was initially well received by reviewers, remarking the new gradient estimator, a new dropbits technique and an interesting observations of better performance when the bitwidth is learned. The experimental results also look promising: showing improved training performance and test performance (including on ImageNet with ResNet-18), properties to reduce quantization error of learned weights, possibility to learn number of bits via learning stochastic bit-dropping masks.

A deeper verification of the specific methods proposed however showed principal issues:
- The methods proposed in the paper are not sufficiently justified by verifiable formal arguments. The proposed intuitive explanations are entangled and actually lead to wrong conclusions. In particular a main claim of the paper that the proposed estimator reduces bias and variance of Gumbel-Softmax estimator was shown wrong and was removed in the revision. The remaining claim that the estimator reduces quantization error is also wrong (see below). With these issues, the gradient part of the paper is largely incorrect, which is in a strong discrepancy with good experimental results.

- Other parts of the paper, comprising the remaining technical contributions are not properly positioned with respect to the SOTA and thus are not necessary novel / improving.

The main technical issues were discussed with all reviewers and were either supported or not objected. Therefore, I am confident that the submission has critical problems and must be rejected. I recommend the authors to thoroughly investigate all the raised issues (by all reviewers) before resubmitting to other venues.

# Details

## Gradient

The overclaim of reducing bias and variance / resolving bias/variance tradeoff has been removed in the revision, but the new gradient estimator remains a central innovation proposed. It is however not justified and cannot indeed be regarded as a good estimator:

* The justification argues about the bias of the Gumbel-Softmax sampling distribution, but the proposed estimator does not use a sampling distribution in the forward pass, and thus by design cannot address this problem.

* The backward pass to use gradient in i_max only (Eq. 3) is not based on any justification at all.

* The remaining claimed good property: "to reduce the quantization error" is, according to the definition in sect. 3.4, not a property of a gradient estimator, but of the stochastic relaxation alone. There is an experimental evidence Fig.2 that the estimator _leads_ to lowering the quantization error. This is however in a contradiction with a direct verification of the proposed estimator that was conducted:
The verification inspects gradient in a single variable $x$ and a linear loss function of the quantized variable $\hat x$.  It shows that the gradient is zero at grid points and discontinuously reverses the direction at half-grid points. Because of such zigzagging, *it does not correspond to minimizing the loss function*, i.e. not a reasonable estimator. The grid points, where the gradient vanishes, may correspond to either local minima or to local maxima of the estimator. Which of the two cases occurs depends exclusively on the sign of the incoming gradient from the loss function. For $L(\hat x) = \hat x$ we observe that the negative gradient points towards nearest grid point, but for $L(\hat x) = -\hat x$ it points away from the nearest grid point, i.e. a step would *increase the quantization error*. The implementation of this verification is attached anonymously:
https://colab.research.google.com/drive/1PibzRMXQ-NVZMUdfgTIK0Q5FxUKyxfqI?usp=sharing

* Alternative existing estimators are not sufficiently discussed: e.g. common deterministic STE, as used in quantization papers: to just treat the quantization operation as identity on the backward pass. Estimator used by Shayer et al. (2018),  Ajanthan et al. 2019 “Mirror descent view for neural network quantization”, Unbiased estimators (e.g. Yin et al. 2019 “ARSM: Augment-REINFORCE-Swap-Merge Estimator for Gradient Backpropagation Through Categorical Variables”). While unbiased estimators may still have too high variance and or be too computationally demanding for deep networks, they can be used for verification purposes.

* The claim that it is not possible to apply unbiased estimators, in particular score function estimator, because of dependency on x is incorrect. See e.g. Schulman et al. 2015 “Gradient Estimation Using Stochastic Computation Graphs”. Many works on advanced unbiased estimators also demonstrate experiments with 2 or more layers of hidden discrete stochastic variables. From this and technical discussion with authors, it is seen that the experimental study is Sec 3.4 is very limited and erroneous.

* The rule by which the probability mass of the dropped bits is uniformly spread over the remaining bits is not justified and appears methodologically incorrect. In Fig.4 it is not clear what bits were dropped and why the mass at $-2\alpha$ has decreased.

## Gradient and Other Techniques relative to SOTA

* The bias problem of GS estimator, detailed in Fig.1. is not novel to me, it is in fact known that the mean under the concrete distribution (of linear or non-linear objective) differs from the mean under the categorical distribution, see e.g.

Lorberbom et al. (2018) Direct Optimization through argmax for Discrete Variational Auto-Encoder (Fig.1)

Andriyash et al. (2018) Improved Gradient-Based Optimization Over Discrete Distributions

Thus analysis of individual samples in Fig.1 appears unnecessary detailed. The issue that the relaxed distribution of Gumbel-Softmax may cause a large estimation error for gradients downstream is already discussed by Louizos et al. (2018) and other works, e.g.

Choi 2017, "Unsupervised Learning of Task-Specific Tree Structures with Tree-LSTMs" Sec 3.2

and Andriyash (2018). This later problem was previously addressed in many cases by the ST Gumbel-Softmax heuristic. This heuristic indeed performs better in CIFAR-10 experiments in the submission / Louizos (2018), which is likely to be a better tuned and more controlled experiment than ImageNet.
* More methods should be discussed that reduce the quantization error during learning. E.g.

Cong et al. (2018): “Extremely low bit neural network: Squeeze the last bit out with ADMM”,

who include terms explicitly minimizing the
quantization error. In fact most works quantizing network weights primarily focus on reducing the quantization error, e.g.

Nagel et al. (2019)" Data-Free Quantization Through Weight Equalization and Bias Correction

* The prior works on learning bit width should be more extensively discussed / compared to, especially if this part becomes central to the submission. E.g.

Baalen et al. (2020) “Bayesian Bits: Unifying Quantization and Pruning” (or references therein if this is considered contemporaneous).

Courbariaux & David (2015): Training deep neural networks with low precision multiplications

* The new hypothesis for quantization is in fact similar to the effect observed elsewhere that quantizing neural networks progressively leads to better results.  E.g.

Zhou et al. (2017) Incremental Network Quantization- Towards Lossless CNNs with Low-Precision Weights.

It is questionable whether the link to the lottery ticket hypothesis is justified, since the latter shows quite the opposite, as reviewers have pointed.